

# Secondary organic aerosol formation from ambient air in an oxidation flow reactor in central Amazonia

Brett B. Palm[1,2], Suzane S. de Sá[3], Douglas A. Day[1,2], Pedro Campuzano-Jost[1,2], Weiwei Hu[1,2], Roger Seco[4], Steven J. Sjostedt[5], Jeong-Hoo Park[6,&], Alex B. Guenther[4,7], Saewung Kim[4], Joel Brito[8,@], Florian Wurm[8], Paulo Artaxo[8], Ryan Thalman[9,^], Jian Wang[9], Lindsay D. Yee[10], Rebecca Wernis[11], Gabriel Isaacman-VanWertz[10,#], Allen H. Goldstein[10,11], Yingjun Liu[3,*], Stephen R. Springston[9], Rodrigo Souza[12], Matt K. Newburn[13], M. Lizabeth Alexander[13], Scot T. Martin[3,14], Jose L. Jimenez[1,2]

[1]Cooperative Institute for Research in Environmental Sciences, University of Colorado, Boulder, USA
[2]Dept. of Chemistry, University of Colorado, Boulder, USA
[3]School of Engineering and Applied Sciences, Harvard University, Cambridge, MA, USA
[4]Dept. of Earth System Science, University of California, Irvine, USA
[5]Earth System Research Laboratory, National Oceanic and Atmospheric Administration, Boulder, CO, USA
[6]National Center for Atmospheric Research, Boulder, CO, USA
[7]Div. of Atmospheric Sciences & Global Change, Pacific Northwest National Laboratory, Richland, WA, USA
[8]Institute of Physics, University of São Paulo, Brazil
[9]Environmental and Climate Sciences Department, Brookhaven National Laboratory, Upton, NY, USA
[10]Department of Environmental Science, Policy, and Management, University of California, Berkeley, CA, USA
[11]Department of Civil and Environmental Engineering, University of California, Berkeley, CA, USA
[12]University of the State of Amazonas, Manaus, Brazil
[13]Environmental and Molecular Sciences Laboratory, Pacific Northwest National Laboratory, Richland, WA, USA
[14]Department of Earth and Planetary Sciences, Harvard University, Cambridge, MA, USA
[&]Now at: Climate and Air Quality Research Department, National Institute of Environmental Research (NIER), Incheon, 22689, Republic of Korea
[@]Now at: Laboratory for Meteorological Physics (LaMP), Université Clermont Auvergne, F-63000 Clermont-Ferrand, France.
[^]Now at Department of Chemistry, Snow College, Richfield, UT, USA
[#]Now at Department of Civil and Environmental Engineering, Virginia Tech, USA
[*]Now at Department of Environmental Science, Policy, and Management, University of California, Berkeley, USA

**Abstract:**

Secondary organic aerosol (SOA) formation from ambient air was studied using an oxidation flow reactor (OFR) coupled to an aerosol mass spectrometer (AMS) during both the wet and dry seasons at the Observations and Modeling of the Green Ocean Amazon (GoAmazon2014/5) field campaign. Measurements were made at two sites downwind of the city of Manaus, Brazil. Ambient air was oxidized in the OFR using variable concentrations of either OH or $O_3$, over ranges from hours to days ($O_3$) or weeks (OH) of equivalent atmospheric aging. The amount of SOA formed in the OFR ranged from 0 to as much as 10 µg m$^{-3}$, depending on the amount of SOA precursor gases in ambient air. Typically, more SOA was formed during nighttime than daytime, and more from OH than from $O_3$ oxidation. SOA yields of individual organic precursors under OFR conditions were measured by standard addition into ambient air, and confirmed to be consistent with published environmental chamber-derived SOA yields. Positive matrix factorization of organic aerosol (OA) after OH oxidation showed formation of typical oxidized OA factors and a loss of primary OA factors as OH aging increased. After OH oxidation in the OFR, the hygroscopicity of the OA increased with increasing elemental O:C up to O:C~1.0, and then decreased as O:C increased further. Possible reasons for this decrease are discussed. The measured SOA formation was compared to the amount predicted from the concentrations of measured ambient SOA precursors and their SOA yields. While measured ambient precursors were sufficient to explain the amount of SOA formed from $O_3$,



they could only explain 10–50% of the SOA formed from OH. This is consistent with previous OFR studies which showed that typically unmeasured semivolatile and intermediate volatility gases (that tend to lack C=C bonds) are present in ambient air and can explain such additional SOA formation. To investigate the sources of the unmeasured SOA-forming gases during this campaign, multilinear regression analysis was performed between measured SOA formation and the concentration of gas-phase tracers representing different precursor sources. The majority of SOA-forming gases present during both seasons were of biogenic origin. Urban sources also contributed substantially in both seasons, while biomass burning sources were more important during the dry season. This study enables a better understanding of SOA formation in environments with diverse emission sources.



# 1       Introduction

Atmospheric submicron aerosols have impacts on radiative climate forcing, air quality, and human health (Pope and Dockery, 2006; IPCC, 2013). Organic aerosol (OA), in particular secondary OA (SOA) formed through various gas-to-particle processes, comprises the majority of ambient submicron particulate mass (Zhang et al., 2007;

Jimenez et al., 2009). SOA can be produced from gases emitted from biogenic, urban, and biomass burning sources, upon oxidation by OH, $O_3$, and $NO_3$ (Ziemann and Atkinson, 2012). In order to mitigate aerosol impacts, the sources, formation, properties, and loss processes of SOA need to be understood, and their uncertainties addressed.

These uncertainties are due in part to limitations in our ability to speciate and quantify the majority of organic

compounds in the atmosphere (Goldstein and Galbally, 2007). These organic compounds range over greater than ten orders of magnitude in volatility, a property which is vital in determining a compound's phase state, lifetime, and fate in the atmosphere (e.g., Donahue et al., 2013). The most volatile organics are called volatile organic compounds (VOCs) and are found almost exclusively in the gas phase, while the lowest volatility compounds are found almost entirely in the particle phase as OA. Under most conditions, VOCs and OA are

generally easier to quantify and speciate. The compounds with volatilities between VOCs and OA (i.e., with saturation vapor concentrations from approximately 1 to $10^6$ µg m$^{-3}$) include semi- and intermediate volatility organic compounds (SVOCs and IVOCs, or S/IVOCs; Robinson et al., 2007), which are more difficult to quantify and speciate. There have been recent attempts to quantify bulk S/IVOCs (Cross et al., 2013; Hunter et al., 2017), to speciate subsets of S/IVOCs (e.g., Zhao et al., 2014; Chan et al., 2016), and to model SOA formation from

anthropogenic S/IVOCs from urban or aircraft emissions (Robinson et al., 2007; Dzepina et al., 2009; Hodzic et al., 2010; Jathar et al., 2011; Miracolo et al., 2011; Woody et al., 2015). The importance of biogenic S/IVOCs for SOA formation in ambient air was also recently demonstrated for the first time (Palm et al., 2016, 2017). However, much remains to be learned about these compounds in order to adequately understand SOA formation on local, regional, and global scales.

Modeling of OA remains extremely uncertain due to uncertainties in these underlying processes (Tsigaridis et al., 2014). SOA parameterizations in atmospheric models have been developed by measuring SOA yields after the oxidation of VOC precursors in large environmental chambers. However, the interpretation and quantification of chamber experiments can be impaired as the result of substantial losses of S/IVOC gases



(Matsunaga and Ziemann, 2010; Zhang et al., 2014; Krechmer et al., 2015; La et al., 2016; Nah et al., 2016) and particles (Crump and Seinfeld, 1981; McMurry and Rader, 1985; Pierce et al., 2008) to the chamber walls. Due to the frequently poor performance of SOA models for field studies (e.g., Tsigaridis et al., 2014), it is of high interest to study SOA formation from ambient air.

Recently, a method of studying SOA formation, namely oxidation flow reactors (OFRs), has been developed. OFRs are relatively small (on the order of 10 L volume) vessels that employ high oxidant concentrations (OH, $O_3$, or $NO_3$) with a short residence time of several minutes (Kang et al., 2007; Lambe et al., 2011a). This technique can achieve anywhere between hours and months of equivalent atmospheric oxidation in an experimental setup that is small and portable. This is in contrast to large Teflon chambers, which are challenging to use for

aging ambient air due to their size and complexity as well as low time resolution (~1 experiment per day). Consequently, such large chambers have been restricted mainly to the aging of exhaust from various emission sources (Presto et al., 2011; Platt et al., 2013). To our knowledge, only one study has used a large chamber to process ambient air for aerosol aging research (Peng et al., 2016a), with no published results on SOA formation from ambient air . In OFRs, ambient air is directly sampled and oxidized in near real-time, allowing rapid

tracking of changes in ambient SOA precursor gases.

OFRs have recently been used to study SOA formation from ambient air in several locations. Oxidation of ambient forest air dominated by biogenic emissions (Palm et al., 2016) and urban air dominated by urban emissions (Ortega et al., 2016) has shown that ambient S/IVOCs are likely important precursors for ambient SOA formation from OH oxidation, but not for $O_3$ or $NO_3$ oxidation (Palm et al., 2017). In contrast to those locations,

the atmosphere in the Central Amazon forest (downwind of Manaus) is influenced by mixed biogenic, urban, and biomass burning sources of SOA precursor gases (Martin et al., 2010; Kourtchev et al., 2016), providing a unique opportunity to study the influence of anthropogenic activities on the atmosphere.

In this work, we use an OFR to investigate SOA formation from the oxidation of ambient air at a tropical rainforest site with varying degrees of urban and biomass burning influence during the GoAmazon2014/5 field

campaign. Ambient air was oxidized by either OH or $O_3$, and the subsequent SOA formation was used to investigate the types, amounts, and diurnal/seasonal changes in the relative contributions of precursor gases to the SOA formation potential of ambient air. SOA yields in the OFR under standard OFR experimental conditions were investigated by injecting and oxidizing known amounts of individual precursor gases in ambient air in the



OFR. These results are discussed in the context of improving our understanding of atmospheric SOA formation and sources.

## 2        Experimental methods

### 2.1        GoAmazon2014/5 field campaign

The Observations and Modeling of the Green Ocean Amazon (GoAmazon2014/5) field campaign took place near the city of Manaus in the state of Amazonas, Brazil, during 2014 and 2015 (Martin et al., 2016, 2017). The majority of the measurements presented in this work were conducted at the "T3" supersite, located approximately 70 km west (downwind) of Manaus, a city of 2 million people. The site was located in a large clearing (2.5 km by 2 km) and surrounded by rainforest, 10 km NE of the town of Manacapuru. These

measurements were taken during the two intensive operating periods, referred to as IOP1 (Feb. 1–Mar. 31, 2014) and IOP2 (Aug. 15–Oct. 15, 2014). IOP1 took place during the wet season, while IOP2 was during the dry season. Measurements were also conducted at the "T2" site, located approximately 10 km west of Manaus on the opposite bank of the Rio Negro, between Mar. 30–May 9, 2014 (wet season) and August 3–September 2, 2014 (dry season). At the T3 site, the wet season was characterized by a total of 705 mm of rainfall, a daily

average temperature of 26°C, and daily average RH of 92%. The dry season at the T3 site was characterized by a total of 243 mm of rainfall, and averages of 27°C and 87% RH. Further details about the GoAmazon2014/5 field campaign can be found in Martin et al. (2016, 2017). Separate studies focusing on ambient aerosol measurements, which are also relevant to this work, are presented in de Sá et al. (2017a, 2017b).

### 2.2        Oxidation flow reactor

The specific OFR used in this work was a Potential Aerosol Mass (PAM) reactor (Kang et al., 2007; Lambe et al., 2011a). The PAM reactor is a cylindrical aluminum tube with a volume of approximately 13 L. Ambient air was sampled through an approximately 2-cm-diameter hole in the inlet plate on one end of the OFR, followed immediately by passing through a coarse mesh grid (1.2 mm spacing) that was coated by an inert silicon coating (Sulfinert by SilcoTek, Bellefonte, PA) in order to minimize gas and particle losses. Two identical OFRs were

located at a height of approximately 4 m above the ground on the roof of a trailer where the instrumentation was located (see Fig. S1). The OFRs were operated at ambient RH and temperature, with a residence time between 2.5–3.9 min. To investigate OH oxidation in the OFR, OH radicals were produced in situ using the





"OFR185" method described elsewhere (Li et al., 2015; Peng et al., 2015). OH exposure ($OH_{exp}$) was estimated using a kinetic model-derived estimation equation, which was discussed in Peng et al. (2015) and can be downloaded from the PAM Wiki (Lambe and Jimenez, 2017). The equation uses inputs of ambient water vapor concentration, temperature, $O_3$ produced in the OFR (measured in the output flow), and external OH reactivity

($OHR_{ext}$) as input parameters (Li et al., 2015; Peng et al., 2015). $OHR_{ext}$ is the OHR from ambient gases such as VOCs. Since there were no direct $OHR_{ext}$ measurements at the T3 site during this campaign, $OHR_{ext}$ was assumed to be equal to the average diurnal profile of measurements from the nearby "T0a" site in Williams et al. (2016), which ranged from 27–74 $s^{-1}$ (shown in Fig. S2). Those measurements were made several meters above the Amazon forest canopy, and were similar to measurements of OHR in other tropical forest locations (Sinha et al.,

2008; Edwards et al., 2013). If the true OHR at the site was different from the average in Williams et al. (2016), the model-estimated $OH_{exp}$ could be different by no more than a factor of 2. The model-estimated $OH_{exp}$ was evaluated by comparing it with measured decay of ambient VOCs and CO (which was injected into the OFR), as shown in Sect. 3.1. For comparison with previous work (e.g., Palm et al., 2016), $OH_{exp}$ was converted to equivalent (eq.) days of atmospheric aging by dividing by a typical 24 h average atmospheric concentration of

$1.5 \times 10^6$ molec $cm^{-3}$ OH (Mao et al., 2009). This eq. age can be scaled accordingly to use other average atmospheric OH concentrations.

To study $O_3$ oxidation, $O_3$ was injected into the OFR using a technique previously described in Palm et al. (2017). Elevated $O_3$ concentrations from hundreds of ppb up to 150 ppm were achieved in the OFR by flowing 0.5 lpm of ultra-high purity $O_2$ (g) over UV lamps (externally to the reactor). The $O_2$ was photolyzed by 185 nm light,

which produced $O(^3P)$ that further reacted with $O_2$ to produce an $O_2+O_3$ mixture. The oxidant flow was then injected through four ports located around the inlet plate inside the OFR. $O_3$ concentrations in the OFR were cycled by adjusting the UV lamp intensity used for $O_3$ production. $O_3$ exposure was calculated by multiplying the $O_3$ concentration by the average residence time in the OFR. This $O_3$ exposure was converted to eq. atmospheric days of oxidation by dividing by a typical 24 h average ambient $O_3$ concentration of 30 ppb. As with OH, this eq.

age of $O_3$ oxidation can be scaled accordingly to apply a different average ambient $O_3$ concentration. Measurements of $O_3$ in the outflow of each OFR were made using a 2B Technologies Model 205 Ozone Monitor and a Thermo Scientific Model 49i Ozone Analyzer at a time resolution of 10 seconds.



In laboratory studies after the campaign, the possible effect of electrical charging by the UV lights on new particle formation dynamics in the OFR was investigated and ruled out (see Sect. S1 and Fig. S3).

### 2.3    Gas and particle measurements

For the measurements at the T3 site, particles in ambient air and after OFR oxidation were sampled using an

Aerodyne high-resolution time-of-flight aerosol mass spectrometer (HR-ToF-AMS, hereafter referred to as AMS; DeCarlo et al., 2006; Canagaratna et al., 2007; de Sá et al., 2017b) and a TSI 3936 Scanning Mobility Particle Sizer (SMPS). Ambient and OFR-oxidized VOC concentrations were sampled during the entire campaign using an IONICON proton-transfer-reaction time-of-flight mass spectrometer (PTR-TOF-MS; Jordan et al., 2009a, 2009b; Müller et al., 2013), which sampled using $H_3O^+$ as the reagent ion during IOP1 and $NO^+$ as the reagent ion during

IOP2. At the T2 site, the gases and particles in ambient air and after the OFR were sampled using an Aerodyne Aerosol Chemical Speciation Monitor (ACSM; Ng et al., 2011) and a unit-resolution quadrupole PTR-MS (IONICON; Lindinger et al., 1998). Additionally, the analysis herein uses concentrations of SQT and several biomass burning tracers, which were measured in ambient air using the semi-volatile thermal desorption aerosol gas chromatograph (SV-TAG; Williams et al., 2006; Zhao et al., 2013; Isaacman et al., 2014).

Measurement details for the SV-TAG during GoAmazon2014/5 can be found in Yee et al. (2017).

At both sites, a system of automated valves (Aerodyne AutoValve) cycled by custom LabVIEW (National Instruments, Inc.) software was used to alternate sampling between ambient and oxidized air. The flowrate through all sampling lines and the OFRs was kept constant at all times by pulling a bypass flow when not actively sampling with a given instrument. Ambient temperature and humidity were recorded using Vaisala HM70

probes. All aerosol samples were dried to below approximately 30% RH prior to or at the same time as being sampled into the measurement trailer, to prevent condensation in the sampling lines when sampling into air conditioned trailers. The decay of injected CO (~2 ppm in reactor) was used to help estimate $OH_{exp}$ in the OFR. CO was measured in ambient air and after oxidation using a Picarro G2401 $CO/CO_2/CH_4/H_2O$ Cavity Ringdown Spectrometer.

OH and $O_3$ oxidation was typically performed in one of two ways. The majority of the time, the oxidant concentration was cycled through ~20 min steps (16–24 min in practice) covering a range of concentrations from no added oxidant to maximum added oxidant over the course of a 2–3 h full cycle. The OFR aerosol was



sampled for the last 4 min of every step, allowing time for the OFR conditions to stabilize before measurement. An alternative method was also used, where the oxidant concentrations were held constant. In this manner, the OA enhancement from a constant amount of oxidation could be sampled every 16–24 min or faster rather than once every 2–3 h. For example, the concentration that typically produces the maximum amount of SOA

formation could be sampled, or the UV lights could be set to achieve the highest oxidant concentrations in order to investigate heterogeneous oxidation.

The aerosol data at the T3 site was corrected for diffusive particle losses in the sampling line (an average correction of 3%) estimated using the Max Planck Particle Loss Calculator (von der Weiden et al., 2009). To account for particle losses to the internal surfaces of the OFR, the OFR data was corrected by the ratio of

ambient OA to the OA measured through the OFR in the absence of added oxidant (an average correction of +6%). A key data product in this work is OA enhancement, which is defined as the OA concentration measured after oxidation minus the ambient OA concentration (measured immediately before and after OFR sampling). The maximum OA enhancement (or maximum SOA formation) observed in this study was typically between 0.5 and 2 eq. days of OH aging, or above 1 eq. day of $O_3$ aging. Unless otherwise specified, the OA enhancements

were corrected for low-volatility organic compound (LVOC) fate to account for losses of condensable gases on OFR surfaces, excessive gas-phase oxidation leading to fragmentation prior to condensation, and limited timescales for condensation in the OFR that are not expected in the atmosphere, as explained in Palm et al. (2016). For completeness, the parameterization for the coefficient of eddy diffusion ($k_e$) as a function of chamber volume (originally used in the LVOC fate correction in Palm et al. 2016) is shown in Fig. S4. The AMS

data at T3 was calculated using a collection efficiency (CE) of 1 for IOP1, as reported in de Sá et al. (2017b), and a composition-dependent CE (mostly 0.5; Middlebrook et al., 2012) for IOP2. These values were verified based on comparison with the SMPS data, which is shown in Fig. S5. The CE of 1 during the wet season, while unusual, corresponds to the value determined during a previous campaign in the wet season in central Amazonia, which is dominated by liquid biogenic SOA under high humidity conditions (Chen et al., 2009; Pöschl et al., 2010;

Bateman et al., 2015).

### 2.4    SOA yields in the OFR measured using VOC standard addition

As presented in Sect. 3.4, SOA yields from the OH or $O_3$ oxidation of several VOCs (and IVOCs in the case of the SQT) were measured in the OFR during GoAmazon2014/5. Yields were measured for $\beta$-caryophyllene (Sigma-



Aldrich, ≥98.5%), (+)-longifolene (Sigma-Aldrich, ≥98%), D-limonene (Sigma-Aldrich, 97%), $\beta$-pinene (Sigma-Aldrich, 99%), $\alpha$-pinene (Sigma-Aldrich, 98%), toluene (Fisher Scientific, 99.8%), and isoprene (Sigma-Aldrich, 99%). The VOCs were injected one at a time in a 20–40 sccm flow of zero air. The liquid VOC standards were contained in a Teflon reservoir which was connected through tee to the zero air flow, such that the VOC

diffused into the air just prior to entering the OFR. This air flowed into the front of the OFR through the same four ports through which $O_3$ was injected for $O_3$ oxidation of ambient air. When $O_3$ oxidation of injected VOCs was performed, $O_3$ was injected through two ports and the VOC was injected through the other two.

The SOA yields were calculated as the mass concentration of SOA formed divided by the mass concentration of the injected VOC that reacted in the OFR. This assumes that the only gas that formed SOA was the injected VOC,

i.e., that there were no SOA precursor gases present in the ambient air (or that they formed an insignificant amount of SOA). The standard addition experiments were performed during daytime hours, when this assumption was valid, with few exceptions. The toluene injection experiment was performed during the evening hours. Concurrently and immediately adjacent to the OFR with toluene injection, a second OFR was operated using OH oxidation of ambient air. In this OFR, approximately 3 $\mu g\ m^{-3}$ of SOA was formed from ambient

precursors during the time of the toluene injection and at a similar $OH_{exp}$, so this amount was subtracted from the amount formed in the toluene-injected reactor to determine the SOA yield from toluene. The OH oxidation of limonene was performed overnight. However, the adjacent OFR was not sampling in a manner that could be used to determine the SOA forming potential of ambient air. Instead, an average value of 5 $\mu g\ m^{-3}$ of SOA (a typical value during the dry season) was assumed to form from ambient precursors and was subtracted when

calculating the SOA yield. Therefore, the measured SOA yield for limonene+OH (presented in Sect. 3.4) is more uncertain than the other measured yields. If the ambient air was assumed to have no SOA precursor gases (very unlikely), then the SOA yield for limonene+OH would be 59% as an upper limit, a value still too low to change the conclusions of these measured vs. predicted SOA analyses.

The isoprene+OH experiment has the caveat that in order to achieve a measureable amount of SOA formation

from isoprene oxidation, approximately 85 ppb of isoprene was injected. This amounted to an added external OH reactivity of approximately 212 $s^{-1}$, which could have resulted in lower $OH_{exp}$ (due to OH suppression, illustrated in Fig. S6) and thus non-OH reactions becoming more important. Regardless, the isoprene injection experiments (including at lower isoprene concentrations) showed that the SOA yield from isoprene+OH



presented in Sect. 3.4 could not be larger than several percent (but was larger than zero). The SOA yields of the SQT species were also more uncertain because the sensitivity of SQT in the PTR-TOF-MS was not calibrated during the campaign. Instead, the PTR-TOF-MS signal for SQT (at $m/z$ 204 when sampling with $NO^+$ reagent ion) was calibrated by comparing the SQT measured in ambient air by the PTR-TOF-MS with the sum of SQT

measured in ambient air by the SV-TAG (shown in Fig. S7) and then using this calibration for the standard addition experiments. The 25% uncertainty of the slope in this calibration directly contributes 25% uncertainty in the calculated SQT yield. While the resulting SQT measurements have significant uncertainty, these measurements nevertheless provide two useful constraints, indicating that SOA yields from SQT in the OFR are not drastically different from chamber-derived yields, and that primary SQTs are a minor contributor to SOA

formation from ambient air in the OFR. Furthermore, variance in the sensitivities of different species of MT and SQT was not accounted for, and will add a small amount of uncertainty.

## 2.5    Predicting SOA formation in the OFR

In Sect. 3.5 below, the measured SOA formation in the OFR is compared with the amount predicted to form in order to investigate which ambient gases are contributing to SOA formation. In order to predict the amount of

SOA that will form, SOA yields are applied to the mass concentrations of all known SOA precursor gases (VOCs and some IVOCs) measured in ambient air.

For OH oxidation, these gases include isoprene, monoterpenes (MT), sesquiterpenes (SQT), benzene, toluene, C8-aromatics (hereafter called xylenes), C9-aromatics (hereafter called trimethylbenzenes), and the sum of four biomass burning tracers (syringol, measured dominantly in the gas phase; vanillin, vanillic acid, and guaiacol,

measured in both gas and particle phases). The SQT and biomass burning tracers were measured using the SV-TAG, and the rest were measured by one of two PTR-TOF-MS instruments sampling at the T3 site (Liu et al., 2016; Martin et al., 2016). First, the fraction of the ambient gas predicted to react in the OFR for a given oxidant exposure was calculated. Then, the OA concentration-dependent SOA yield parameterizations from Tsimpidi et al. (2010) were used to calculate the amount of SOA predicted to form (except for isoprene, where the yield

parameterization from Henze and Seinfeld (2006) was used).The average yields used in these calculations for wet(dry) season were 3%(5%) for isoprene, 10%(18%) for MT (also used for the biomass burning tracers), 10%(23%) for SQT, 11(22%) for benzene and toluene, and 12%(26%) for xylenes and trimethylbenzenes. These yields were calculated at the average ambient OA concentrations of 1.3 µg m$^{-3}$ and 9.5 µg m$^{-3}$ at T3 in the wet



and dry seasons, respectively. The SOA yields include absorptive partitioning, where the SOA yields increase with increasing OA concentrations. To test whether absorptive partitioning was occurring in the OFR, the dependence of the maximum SOA formation measured from OH oxidation during the dry season on the ambient OA concentrations was investigated. As shown in Fig. S8, absorptive partitioning is likely playing some role, but may not have as strong of an effect as suggested in the published SOA yields used above.

For $O_3$ oxidation, ambient MT and SQT were used to predict SOA formation. Other VOCs, e.g. the aromatic compounds mentioned above that were used in the OH oxidation analysis, were not included in the $O_3$ oxidation analysis because such compounds lack non-aromatic C=C bonds and tend to be non-reactive towards $O_3$ for the concentrations used in this study (Atkinson and Arey, 2003). As for previous $O_3$ oxidation experiments in an OFR, representative SOA yields of 15% for MT and 30% for SQT were used (Palm et al., 2017). Due to the uncertainty in these yields, the lack of speciation of MT at the T3 site, and the lack of published yields for the variety of SQT that were speciated by the SV-TAG, these yield values were chosen to be generally representative of published values (e.g., Jaoui et al., 2003, 2013; Ng et al., 2006; Pathak et al., 2007, 2008; Shilling et al., 2008; Winterhalter et al., 2009; Chen et al., 2012; Tasoglou and Pandis, 2015; Zhao et al., 2015).

## 3    Results and Discussion

### 3.1    Using VOC decay to determine OH and $O_3$ exposure

One of the benefits of the OFR system over environmental chambers is the ability to rapidly change the amount of oxidant in the OFR over a wide range of concentrations. As described above, $OH_{exp}$ in the OFR was estimated using a model-derived equation, while $O_3$ exposure was estimated as the measured $O_3$ concentration after the OFR multiplied by the average residence time. Because there are uncertainties related to these estimates (e.g., uncertain OH reactivity, residence time distribution, intrinsic uncertainties of the model and estimation equations), it is important to use in situ measurements to verify the exposures achieved in the OFR. This can be done by measuring the decay of various gases, including gases present in ambient air or gases that are injected into the OFR. Previous experiments have injected deuterated compounds, which prevent contamination of the signal with ambient gases and allow the reaction rate constant of the injected compound to be known precisely (Bruns et al., 2015). In this work, decay of ambient toluene and MT and injected CO was used to verify the OH and $O_3$ exposures. Any changes in the ambient concentrations of these gases between the times of the



surrounding ambient measurements and the time of the decay in the OFR (approximately 5 min apart) translates into noise in the measurement of the fraction reacted. The speciation of MT in ambient air was also unknown. In this analysis, the fraction remaining was predicted using $\alpha$-pinene (an important MT in the Amazon; e.g., Rinne et al., 2002; Jardine et al., 2015) with rate constants $k_{OH} = 5.3 \times 10^{-11}$ and $k_{O_3} = 8.6 \times 10^{-17}$ cm$^3$ molec$^{-1}$ s$^{-1}$ (Atkinson and Arey, 2003).

The decay of ambient MT, ambient toluene, and injected CO in the OH-OFR is shown in Fig. 1, along with the theoretical decay curves predicted assuming either plug flow (i.e., a single residence time) or using the residence time distribution (RTD) for particles from Lambe et al (2011a), which is likely to be more skewed away from laminar flow than the RTD in this work (due to our use of a larger inlet). In general, the OH$_{exp}$ predicted from the model-derived equation matches the OH$_{exp}$ estimated from the decay of gases within a factor of approximately 2-3, consistent with expectations (Li et al., 2015). The model equation appears to over-predict OH$_{exp}$ at the lowest achieved exposures for MT (but not for toluene or CO), while under-predicting at the highest exposures for CO while over-predicting for toluene. There could be several reasons for these differences. The speciation of MT in ambient air is likely to change with time and the combined MT signal is likely to decay at a different rate than the assumed $\alpha$-pinene rate. Interferences in the PTRMS signal for MT due to oxidation products may mask the decay of these species at low remaining fraction. Also, it is likely that the true RTD has some differences from the one used in the calculation, and perhaps some variability in time. If even small plumes of ambient air transit through the OFR without being exposed to as much oxidant due to variability in the internal air flow fields, this can lead to increases in the measured fraction remaining, particularly for lower exposures. At high exposures, the model assumes that the OH reactivity of the ambient air decays during OFR transit at the same rate as the reaction of SO$_2$ (Peng et al., 2015). If this estimated decay is too slow (e.g. due to faster decay of isoprene-related reactivity), it could lead to an under-prediction of OH$_{exp}$ at high exposures.

The decay of MT in the O$_3$-OFR is shown in Fig. 2, along with predictions for the plug flow and Lambe et al. (2011a) residence times. Again, the O$_3$ exposures estimated from the model and from MT decay match within a factor of approximately 2-3. All MT were reacted after an exposure of 1 eq. day.

### 3.2    Examples of SOA formation from ambient biogenic and urban gases in OFR



A basic premise of the OFR technique (as used in this work) is that SOA precursor gases entering the OFR can be oxidized to form SOA. A simple way to investigate and illustrate this concept for ambient experiments is to compare SOA formation with ambient VOCs over a period of time. Fig. 3 shows a two-night example of OA measured in ambient air compared to OA measured after OH oxidation at the T3 site, along with ambient total

MT and copaene (a SQT). In this example, the $OH_{exp}$ was kept nearly constant for the entire time at approximately 3 eq. days, near the range where maximum SOA formation was usually observed. Using this method, maximum SOA formation was sampled every 24 min rather than every 2–3 h as with the standard cycling of $OH_{exp}$. Note that in theory, ambient and OFR measurements could be alternated at much faster frequencies (as fast as ~10 s). In practice during this time period, the instrumentation was alternating between

measurements of ambient air, two OFRs, and a thermodenuder, and longer averages of the data (1–2 min) were preferred to reduce noise and data volume, limiting the frequency with which the OFR measurements were taken. In Fig. 3, the times when SOA was formed in the OFR clearly coincide with the spikes in ambient MT and SQT concentrations, illustrating an example of likely biogenic-dominated SOA formation. This is evidence that the SOA being formed in the OFR was derived from gases that were entering the OFR. Importantly, this example

illustrates that the ambient precursor concentrations at the T3 site can change rapidly, even faster than the typical 2–3 h cycles.

Another example of SOA formation from ambient precursors, this time from the T2 site (close to Manaus), is shown in Fig. 4. In this example, the $OH_{exp}$ was cycled through the whole range of eq. ages, including one step each cycle with no OH. In the OFR, SOA was formed at three distinct times, labeled Periods 1, 2, and 3 in Fig. 4.

During Period 1, ambient MT concentrations were near zero, and elevated concentrations of xylenes and TMB strongly suggest the presence of an urban plume affecting the T2 site. The SOA formed during this cycle was likely formed from predominantly urban precursors. In contrast, the SOA formed during Period 3 was produced in the presence of MT but not the urban tracers, suggesting the SOA was predominantly biogenic. The SOA formed during Period 2 in Fig. 4 was produced in the presence of both urban and biogenic gases, and likely was

formed from a mix of both types of gases. These two examples clearly illustrate the usefulness of the OFR technique for measuring potential SOA formation from ambient air.

### 3.3    OA enhancement vs. photochemical age



As part of the GoAmazon2014/5 campaign, the formation of aerosol from the oxidation of ambient air was sampled over a wide range of conditions. These conditions include the changes of ambient air composition between the wet and dry seasons, and the diurnal, synoptic, and other changes during each season. OH oxidation of ambient air was performed at both the T2 and T3 sites, and $O_3$ oxidation was performed at the T3

site only. A basic way to view the differences across these conditions is by comparing the absolute OA enhancement from each subset as a function of photochemical age. This is shown for OH oxidation in Fig. 5 and $O_3$ oxidation in Fig. 6, split into daytime (06:00–18:00 LT) and nighttime (18:00–06:00 LT) for each season and location. The T3 site data are shown both without the LVOC fate correction and with the correction. For OH oxidation, the LVOC fate correction was applied at ages below 10 eq. days only. At higher ages, heterogeneous

oxidation leads to substantial fragmentation/evaporation of preexisting particles. This effect competes in uncertain ways with the condensation of LVOCs, so the LVOC fate correction cannot be applied with confidence. Therefore, the data are shown without the LVOC fate correction in order to illustrate the measurements over the entire age range. Also, the LVOC fate correction was not applied for data from the T2 site, because the requisite measurements of size distribution after oxidation in the OFR were not available. However, it is likely

that the correction would be approximately of the same magnitude as for the T3 data. The LVOC fate correction was not applied for daytime $O_3$ oxidation data because the signal-to-noise of SOA formation was too low.

For OH oxidation at each site and season, an increasing amount of SOA formation was observed for increasing ages, up to a maximum amount of SOA formed in the range of approximately 1–4 eq. days of OH oxidation. At higher ages, the net amount of SOA formed became less or even negative (net loss of OA compared to ambient

air). This result is due to a combination of two effects (which have also been observed previously): rapid oxidation of condensable gases prior to those gases having time to condense on particles, leading to fragmentation in the gas phase that produces volatile oxidation products; and heterogeneous oxidation of preexisting (and newly formed) particle mass, leading to fragmentation and evaporation of the particles (George and Abbatt, 2010; Lambe et al., 2012; Ortega et al., 2016; Palm et al., 2016).

At both sites, a maximum of approximately 4–5 times more SOA was formed from ambient air during the dry season compared to the wet season. During the dry season, the maximum amount of SOA formed at the T2 site during nighttime was about 50% larger than at the T3 site during nighttime (assuming the LVOC fate correction was the same at each site). It may be the case that this increased SOA formation was due to a larger urban



source strength in the closer proximity to the city of Manaus. The maximum amounts of SOA formed at all other times were approximately equivalent at each site. These measurements suggest that the seasonal changes in SOA precursor gases are more important to potential SOA formation than the proximity to Manaus. One possibility is that a substantial fraction of the urban SOA had already formed by the time the air passed over the T2 site, so formation in the OFR of the remaining potential SOA did not lead to a very large difference between the sites sources.

As shown in Fig. 6, approximately 2–3 times more SOA was formed from $O_3$ oxidation during the dry season than the wet season, again with typically higher formation during nighttime than daytime hours. The amount of SOA formation increased with $O_3$ eq. age, with maximum values above 1 eq. day of $O_3$ oxidation. This is consistent with the age at which the ambient MT (and likely other compounds) were all reacted, as shown in Fig. 2. As observed previously at another biogenic site, $O_3$ oxidation of ambient air produced at most ~ $1/6^{th}$ of the SOA that was formed from OH (Palm et al., 2017).

### 3.4 Investigating SOA yields in an OFR using standard addition

One of the original design intents of the PAM OFR was to oxidize air containing aerosol precursors and measure the "potential" amount of aerosol that can be formed. Since the initial development of the PAM reactor (Kang et al., 2007), subsequent research has shown that there are many factors related to exactly how the PAM reactor is operated that can affect the amount of aerosol that is formed (Peng et al., 2015, 2016b; Hu et al., 2016; Palm et al., 2016). For OH oxidation, the amount of SOA formed increases as $OH_{exp}$ increases, up to a maximum amount of SOA formed in the range of $OH_{exp}$ between the exposure where most of the reactive precursor gases have reacted and approximately 5 eq. days of exposure. At higher exposures, the high amounts of OH radicals start reacting many times with gases faster than condensation can occur, which fragments them to form volatile oxidation products that can no longer condense. Also, these high OH exposures start heterogeneously oxidizing any preexisting (or newly formed) aerosol, leading to fragmentation and evaporation (Lambe et al., 2012; Ortega et al., 2016; Palm et al., 2016). So, in order to measure the maximum "potential" aerosol formation, the experiment needs to be operated over the range of exposures below approximately 5 eq. days.



Achieving the proper range of $OH_{exp}$, however, is also non-trivial. $OH_{exp}$ in the OFR has been shown to be sensitive to many factors, including UV photon fluxes, sample air composition, water vapor content, external OH reactivity, and OFR residence time and distribution (Li et al., 2015; Peng et al., 2015). All of these factors need to be considered when estimating $OH_{exp}$. Special care must be taken to avoid operating the OFR at

conditions that lead to significant influence on the chemistry from non-OH reactions (e.g., photolysis; Peng et al., 2016b). Also, Palm et al. (2016) showed that some fraction of the condensable gases will condense on OFR walls, sampling lines, or react further with OH and fragment instead of condensing to form SOA. This behavior is sensitive to the condensational sink (i.e., surface area of seed aerosol) available in the OFR. These alternate fates are artifacts of the OFR experiment, and must be corrected using the measured condensational sink in

order to determine the true potential aerosol mass that would form in the atmosphere.

All of these effects can matter for OFR experiments that attempt to compare measured vs. predicted SOA formation, and they have been considered in, e.g., the SOA formation from oxidation of ambient air in Palm et al. (2016, 2017) and in the subsequent analysis in this work. In these analyses, this carefully-quantified maximum amount of SOA formation was compared to the amount predicted to form from the oxidation of the

speciated precursor gases measured in ambient air. The amount of predicted SOA was estimated by applying typical chamber-derived SOA yields to the measured amount of ambient gas. One important aspect of this analysis that has not been as carefully examined in the literature is whether (or how well) these typical chamber SOA yields apply to the SOA formation in the OFR, particularly under ambient operating conditions. Several previous results have suggested that SOA yields in the OFR were similar to published chamber yields (Kang et

al., 2007; Bruns et al., 2015; Lambe et al., 2015). However, these conclusions were often drawn from experiments that likely suffered from one or more of the following issues: (1) not considering factors such as high VOC concentrations (high external OH reactivity, leading to OH suppression) when determining $OH_{exp}$ (Peng et al., 2015); (2) not considering the alternate fates of condensable gases, particularly for short OFR residence times or when using no seed aerosol (Palm et al., 2016); (3) not considering possible non-OH reactions,

particularly under "high risk conditions" such as high external OH reactivity (Peng et al., 2016b); (4) not considering possible effects of the water vapor concentration of the sample air on both $OH_{exp}$ and aerosol liquid water content (Peng et al., 2015; Palm et al., 2016); and (5) not performing the SOA yield experiments at atmospherically relevant OA concentrations.



Due to these possible limitations of prior OFR SOA yield studies, during the GoAmazon2014/5 field campaign we endeavored to investigate whether SOA yields in the OFR are indeed consistent with published chamber yields, while avoiding or at least considering all of the above-mentioned potential pitfalls (see Sect. 2.4 for more details). SOA yields were quantified by injecting several pure VOCs (individually) into the ambient air at the entrance to the OFR, exposing them to varying concentrations of either OH or $O_3$, and measuring the resultant SOA formation as well as VOC decay. By injecting the VOCs into ambient air, we were able to measure the yields at ambient temperature, humidity (and aerosol liquid water content), and seed OA concentrations. The injected VOC concentrations were also kept low in order to minimize the undesired effects of added external OH reactivity (with the exception of isoprene, as discussed in Sect. 2.4 and Fig. S6). Both constant and stepped oxidant concentrations were used in these experiments. The amounts of OH aging used for these yield calculations were all below approximately 5 eq. days of aging, in order to minimize the influence of heterogeneous oxidation and excessive oxidation reactions in the gas phase. Conversely, $O_3$ ages above 1 eq. day were used.

The measured SOA yields are shown in Fig. 7, along with relevant yield parameterizations used in box and chemical transport models (Tsimpidi et al., 2010) using low-$NO_x$ yields (Lane et al., 2008a) corresponding to the expected conditions in the OFR (Li et al., 2015). The SOA yields (listed in Table S1 along with the OA mass concentrations at which they were measured) were measured to be 52% for $\beta$-caryophyllene+OH, 51% for longifolene+OH, 27% for $\beta$-caryophyllene+$O_3$, 30% for limonene+OH, 18% for $\beta$-pinene+OH, 11% for $\alpha$-pinene+OH, 17% for limonene+$O_3$, 21% for $\alpha$-pinene +$O_3$, 11% for toluene+OH, and 6% for isoprene+OH. These yield values are generally consistent (within a factor of 2 for comparable OA mass concentrations) with the values that have been determined in large chambers, with the averages being 0.9, 1.3, 0.5, and 0.9 times the respective chamber-derived yields for MT, SQT, toluene, and isoprene. Importantly, there is no indication that the OFR is *more* efficient at forming SOA than the chamber yields would indicate. This confirms that the OFR can be used to quantitatively determine the amount of SOA that would form upon oxidation of an ambient mix of precursor gases. Furthermore, it supports the analyses presented in Palm et al. (2016, 2017) that ambient VOCs alone could explain the amount of SOA formed from $O_3$ oxidation but not OH oxidation, where unspeciated S/IVOCs contributed a majority of the SOA formation in the OFR.

### 3.5    Measured vs. predicted SOA formation



When SOA precursor gases enter the OFR, either in ambient or injected air as illustrated above, SOA can be produced by oxidizing the gases in the sampled air. As shown in Sect. 3.4, when a known concentration of VOCs is added to the OFR, the amount of SOA formed upon oxidation by either OH or $O_3$ is consistent with what would be expected from published chamber experiments. Therefore, when comparing the measured SOA

formation from the oxidation of ambient air to the amount predicted to form from measured ambient gases, we can determine if all of the SOA formation is accounted for, or if there are other SOA-forming gases present in ambient air that are not being measured and quantified. Previous studies of OFR oxidation of urban or pine forest ambient air has shown that poorly characterized S/IVOCs are likely an important source of SOA from OH oxidation (Ortega et al., 2016; Palm et al., 2016). In contrast, SOA formed from $O_3$ and $NO_3$ oxidation in a

biogenic environment can be accounted for from ambient VOCs alone, indicating that S/IVOC precursors tend not to have C=C double bonds (Palm et al., 2017).

The measured SOA formation (at the eq. ages of maximum SOA production, as discussed above) from ambient air in the OFR during GoAmazon2014/5 is shown in Fig. 8, for both wet (IOP1) and dry (IOP2) seasons and both OH and $O_3$ oxidation and with linear regressions shown. The measured SOA formation is corrected for LVOC

fate. The predicted SOA formation was estimated by applying typical chamber SOA yield values to measured ambient VOC concentrations, as described in Sect. 2.5.

OH oxidation of ambient air produced on average 6.5–8 times more SOA than could be accounted for from ambient VOCs. This is consistent with previous OFR measurements, suggesting that typically unmeasured ambient gases play a substantial role in ambient SOA formation from OH oxidation. The amount of SOA formed

from $O_3$ oxidation was on average similar or slightly larger than the amount that could be explained from measured ambient VOCs. This measurement is noisy (particularly in the dry season, when using a difference measurement to quantify several tenths of μg m$^{-3}$ of SOA formation on top of ~10–20 μg m$^{-3}$ is difficult). Given the uncertainties in e.g. VOC speciation and yield, it is consistent with the previous OFR measurements in a pine forest where ambient VOCs could explain all SOA formation from $O_3$ oxidation (Palm et al., 2017). These non-

VOC ambient gases are likely to be the typically unmeasured/unspeciated class of lower volatility S/IVOCs. However, there were no instruments dedicated to quantifying the total concentration of these gases during GoAmazon2014/5. The measurement of such gases remains a critical gap in our understanding of the lifecycle of carbon in the atmosphere. However, the SOA formed in the OFR that cannot be accounted for by VOCs is





effectively an integrated measure of these S/IVOC gases (multiplied by their SOA yield). They are measured by first converting them into SOA, which is much more readily measurable and quantifiable than S/IVOCs with current instrumentation.

Whereas the slope of the measured vs. predicted SOA formation from pine forest air in Palm et al. (2016) was

roughly constant at approximately 4, the slope of the measured vs. predicted SOA formation from OH oxidation in the Amazon varied as a function of time of day. The diurnal cycles of measured and predicted SOA formation are shown for both seasons in Fig. 9. The predicted SOA was on average slightly lower during nighttime than during daytime. The cycle of measured SOA formation was the opposite, leading to slopes (in Fig. 8) that were lowest during daytime and highest in the hours before sunrise. The reasons for the observed trends are unclear,

but likely result from the confluence of several processes, e.g., diurnal changes in emission and concentration profiles (of VOCs and/or S/IVOCs), boundary layer dynamics, and varying ambient oxidant concentrations.

In addition to showing the diurnal average SOA formation, Fig. 9 also illustrates that a wide range of potential SOA formation is possible at any given time of day. There were some nights when as little as 1 $\mu$g m$^{-3}$ of SOA was formed, and other nights when nearly 10 $\mu$g m$^{-3}$ was formed. During the nights when little SOA was formed,

Fig. 8 shows that these nights also had the lowest predicted amounts of SOA formation. This shows that, while the amount of SOA formation correlated with measured ambient SOA precursor VOCs, they could not quantitatively explain the total amount of SOA formed. Other SOA-forming gases were apparently present at the same times as VOCs, though in varying ratios to those VOCs.

### 3.6     Positive Matrix Factorization (PMF) of SOA after OH oxidation

PMF is a common technique for source apportionment of ambient aerosol (e.g., Ulbrich et al., 2009; Zhang et al., 2011). It can be used to split the full mass spectrum into the sum of several statistical factors, where each factor is the mass spectrum that is produced from a group of related molecules in the ambient aerosol that vary together in time. Here, we present results of PMF analysis of OA after OH oxidation, as an investigation into what types of SOA were formed in the OFR and how heterogeneous oxidation affected the types of pre-existing

OA that entered the reactor in ambient air. In related analyses, the results of PMF analysis for ambient OA (i.e., not oxidized in an OFR) are presented in de Sá et al. (2017a, 2017b). To the best of our knowledge, the results presented in this study are the first report of PMF analysis of the complete OA after oxidation in an OFR.



First, PMF was applied to only the unoxidized measurements through the OFR. The resulting PMF factors were similar to the factors identified in ambient air (de Sá et al., 2017a). The mass concentrations of these unoxidized OFR factors represent the ambient air baseline against which OA enhancements can be calculated. These factor profiles for the wet and dry seasons are shown in Fig. S9–S10. The analysis herein describes how characteristic

factors changed as a function of OH aging in the OFR. The results should be interpreted in the context of how OFR oxidation affects the concentration of these types of factors, which are commonly found in PMF analyses of ambient OA. The interpretation of the factors in ambient OA is outside of the scope of this analysis, and are the subject of separate studies (de Sá et al., 2017a, 2017b).

Several factors that were identified during both wet and dry seasons are common in PMF literature, including

hydrocarbon-like OA (HOA), biomass burning OA (BBOA), isoprene epoxydiols-derived SOA (IEPOX-SOA), and several oxidized OA (OOA) factors that represent SOA (e.g., Aiken et al., 2009; Ulbrich et al., 2009; Zhang et al., 2011; Hu et al., 2015). A factor with a characteristic signal at $m/z$ 91, referred to here as the "Fac91" factor was also identified during the wet season. The HOA and BBOA factors are typically dominated by primary OA (POA, i.e., direct particle emissions), and are not expected to be produced from the chemistry in the OFR. IEPOX-SOA,

while representing a type of SOA, was also not expected to be produced in the OFR. In the atmosphere, IEPOX-SOA is formed via reactive uptake of gas-phase IEPOX onto acidic aerosols (Eddingsaas et al., 2010; Froyd et al., 2010; Surratt et al., 2010; Lin et al., 2012; Liao et al., 2015). As detailed in Hu et al. (2016), IEPOX can be formed in the gas phase in the OFR at accelerated rates, but the rate of reactive uptake in the OFR does not increase with the increased OH concentrations, resulting in negligible formation of IEPOX-SOA in the OFR.

For the wet season, PMF of the OH-aged aerosol was performed with a total of 6 factors, using the Source Finder analysis software (SoFi, version 6.2; Canonaco et al., 2013) to constrain the HOA, BBOA, Fac91, and IEPOX-SOA factors to be exactly the same as the factor profiles found in unoxidized ambient air, and allowing for two free-spectrum SOA-related factors. These two factors are referred to as less-oxidized OOA (LO-OOA) and more-oxidized OOA (MO-OOA) based on their relative O:C. For the dry season, the HOA, BBOA, and IEPOX-

SOA factors were constrained and the two OOA factors were allowed, for a total of 5 factors.

The changes in the mass concentrations associated with each factor after OH oxidation compared to before oxidation are shown for the dry season in Fig. 10. The results during the wet season were generally similar, so they are shown in Fig. S11. The factors associated with POA or with SOA from reactive uptake processes were





not enhanced by the OFR oxidation, as expected, and were depleted as the eq. age of OH oxidation increased. The Fac91 factor also fell into this category. Notably, the factor concentrations decayed at different rates, with HOA (and Fac91) decaying at faster relative rates than IEPOX-SOA and BBOA. This is particularly clear in the dry season. The decay of these factors at higher eq. ages is likely due to heterogeneous oxidation leading to

fragmentation and evaporation of the preexisting aerosol, or conversion of the POA factors into MO-OOA that remains in the particle phase.

In contrast, the OOA factors were produced in the OFR at concentrations that varied as a function of eq. age. SOA associated with the LO-OOA factor was produced at the lower ages, peaking around approximately 1 eq. day of aging. As the age increased, a plateauing of LO-OOA was observed, followed by a decay. Eventually at

ages larger than 6-9 equivalent days a decrease of LO-OOA below the preexisting amount in ambient air was observed, indicating that the pre-existing ambient LO-OOA was being heterogeneously oxidized, and that no new LO-OOA was surviving the OFR (either it was not formed, or it was formed but then converted into MO-OOA or heterogeneously oxidized to gas-phase species). The MO-OOA factor concentration increased as a function of age, peaking and then plateauing around 10 eq. days of aging, where heterogeneous oxidation was a

dominant process affecting the OA sampled out of the OFR.

This PMF analysis shows that the SOA formed in the OFR from hours up to several days of eq. OH aging produces a mass spectrum in the AMS that resembles the spectra of ambient OOA. The mass spectrum of the SOA formed from OH oxidation was correlated ($R^2$=0.72-0.93; shown in Fig. S12) with spectra of the SOA formed from the injected VOCs from the standard injection experiments in Sect. 3.4. These correlations show that the

SOA formed from OH oxidation of ambient air appeared similar to SOA from known precursors, but the spectra from the different precursors appear too similar to be able to differentiate the SOA sources in ambient air from the spectrum alone. The decay of HOA, BBOA, and IEPOX-SOA factors suggest that heterogeneous oxidation is indeed minor at the low eq. ages, though it may have a stronger impact on HOA. For OH oxidation of urban air in an OFR, this should be considered. At the highest ages, this analysis suggests that all of the factors (except

MO-OOA) decay by 70-80% relative to their initial concentration, and that the remaining aerosol is mostly associated with the MO-OOA factor. This suggests that heterogeneous oxidation could be a source of MO-OOA in the atmosphere, particularly in more highly aged particles. Since the oxidation inside the OFR occurs at the same RH as ambient air, this also indicates that diffusion in the ambient OA in the studied region is fast enough,



so that most ambient OA is not shielded from oxidation by slow diffusion. This is consistent with previous measurements showing that regional SOA at this site was in liquid form most of the time under ambient RH (Bateman et al., 2015).

### 3.7 Hygroscopicity of the organic component of CCN after OH oxidation

In addition to characterizing the OA mass as a function of eq. age of oxidation in an OFR, we can also investigate the properties of the OA as a function of aging in the OFR. During Oct. 7–15 in the dry season, the OFR output was size-selected by a DMA and the size selected particles were then measured by a CCN counter and a CPC to derive activated fraction as a function of supersaturation. The hygroscopicity ($\kappa$) of the CCN was determined from the spectrum of activated fraction (Mei et al., 2013; Thalman et al., 2017). When coupled with the

chemical speciation measurements provided by the AMS and using the relatively well known values of $\kappa$ for the inorganic aerosol components, the $\kappa$ of the organic component of CCN ($\kappa_{OA}$) can be determined (Petters and Kreidenweis, 2007). This analysis for ambient OA during GoAmazon2014/5 has been presented elsewhere (Thalman et al., 2017). Here, we present an analysis of how $\kappa_{OA}$ changed upon oxidation in the OFR.

    Due to the sampling time requirements of the CCN counter, these experiments were performed while keeping

the amount of oxidation in the OFR constant. As previous research of OH oxidation in an OFR has illustrated (Lambe et al., 2012; Ortega et al., 2016; Palm et al., 2016, 2017), the OFR can be operated under conditions dominated by SOA formation with limited heterogeneous oxidation (at ages below approximately one to a few eq. days), conditions dominated by heterogeneous oxidation with minimal new SOA formation (the highest ages above approximately 10 eq. days), or conditions where both processes are occurring (the intermediate age

range). When sampling the OFR with the CCN counter during GoAmazon2014/5, the OFR was operated to investigate both the SOA formation and heterogeneous oxidation regimes, at separate times. During nighttime hours, when SOA-forming gases were expected to be present in ambient air in their highest amounts, the OFR was operated at a near constant age in the range of 1–3 eq. days. During daytime hours, when SOA-forming gases were present in lower concentrations, the OFR was operated at a near constant age in the range of 12–44

eq. days of OH aging. To increase confidence that the measurements at the very high eq. ages were a result of heterogeneous oxidation of preexisting aerosol and not influenced by new SOA formation of highly oxidized gases, a parallel-plate carbon filter denuder (Sunset Laboratory Inc.) was mounted on the inlet of the OFR during some of these high-age measurements in order to remove SOA-forming gases from ambient air. For



reference, the evolution of bulk O:C vs. eq. days of OH aging for all data during the dry season is shown in Fig. S13.

Figure 11 shows $\kappa_{org}$ of 160 nm mobility diameter particles as a function of bulk O:C in both ambient and oxidized air. The $\kappa_{OA}$ of ambient OA was in the range of 0.05–0.2, and increased monotonically with increasing

O:C. When operating the OFR in the 1–3 eq. day range (corresponding to OFR data with O:C less than ~1.0), the OH-aged OA maintained the same slope of monotonically increasing $\kappa_{OA}$ with increasing O:C, but the data were shifted to the right (to higher O:C values for a given $\kappa_{OA}$). In other words, the OH oxidation led to an increase in O:C, but the value of $\kappa_{OA}$ increased a smaller amount per unit increase in O:C compared to the rate measured in the slope in the ambient OA, so the trend in the oxidized OA kept the same slope but with a different intercept.

This indicates that the process(es) in ambient air that modulates $\kappa_{OA}$ and O:C is likely not dominated by condensation of new SOA from hours to several days of aging, which was the process specifically studied here. For example, processes such as aqueous chemistry or the formation of IEPOX-SOA through particle phase reactions could contribute substantially to the composition and properties of ambient OA.

The measurements made at high eq. OH ages (corresponding to O:C greater than ~1.2) showed unexpected

results. Instead of continuing to increase at very high O:C values, $\kappa_{OA}$ decreased to below 0.1 with increasing O:C above 1.2, even as O:C increased to higher than 1.4. This trend appears regardless of whether the denuder was used to remove any VOCs. While this decrease in $\kappa_{OA}$ with increasing O:C was unexpected, it is not necessarily inconsistent with previous measurements that have generally shown only increasing $\kappa_{OA}$ with increasing O:C (or $f_{44}$, the fraction of signal found at $m/z$ 44). Those previous measurements involved the heterogeneous oxidation

of POA surrogate particles (Petters et al., 2006a; George et al., 2009; Cappa et al., 2011; Lambe et al., 2011a) and measurements of SOA formed in an OFR in laboratory experiments (Massoli et al., 2010; Lambe et al., 2011a, 2011b). The experiments of heterogeneous oxidation of POA did not achieve O:C values or eq. ages as high as the maximum values achieved in this study (O:C of ~0.25). At their highest amounts of oxidation, a plateau in $\kappa_{OA}$ of approximately 0.1 or lower was observed, which is indeed consistent with the endpoint $\kappa_{OA}$

values achieved at the highest ages in this study. OFR measurements of CCN activity of SOA formed in the OFR in Massoli et al. (2010) and Lambe et al. (2011a, 2011b) did achieve O:C levels and eq. ages closer to the levels in this study, and reported continued monotonic increases in $\kappa_{OA}$ with increasing O:C. However, in those experiments, SOA was formed in the reactor by homogeneous nucleation of gas-phase oxidation products of





injected precursors, and no organic seed aerosol was used. Therefore, the OA measured from the OFR was likely dominated by SOA formed via condensation of highly oxidized gases (with limited time for heterogeneous oxidation to occur after condensation). The gases that condense to form SOA after being oxidized in the gas phase at such high ages (up to 13–20 eq. days in those studies) are likely not representative of the molecules in typical atmospheric particles because of the excessive number of reactions with OH prior to condensation (Palm et al., 2016). The production of OA in those studies stands in contrast to the processing of the OA sampled from the OFR during GoAmazon2014/5. The OA in this study started as real ambient OA, and was dominantly affected either by condensation of oxidation products of atmospherically relevant reactions with OH, or by heterogeneous (or condensed phase) reactions with OH with minimal influence from condensation of gases (especially when using the denuder on the OFR inlet). These results suggest that heterogeneous or particle phase reactions of OA with OH can lead to a decrease in $\kappa_{OA}$.

The specific processes that lead to the observed decrease in $\kappa_{OA}$ due to heterogeneous oxidation are uncertain. One possible process that can lead to a decrease in CCN activity is oligomerization, causing an increase in the molecular weight and decrease in polarity of the particulate organic molecules (VanReken et al., 2005; Petters et al., 2006b; Xu et al., 2014). Oligomerization was suspected in a previous study where heating of OA in a thermodenuder led to a decrease in $\kappa_{OA}$ (Kuwata et al., 2011). Other studies have shown that OH oxidation in the condensed phase can lead to oligomerization (e.g., Altieri et al., 2008; Lim et al., 2010; Sun et al., 2010; Tan et al., 2010, 2012). Similar processes may have occurred in this study.

Another possible reason for the observed decrease in $\kappa_{OA}$ at high O:C could be that heterogeneous oxidation might lead to different results depending on the specific properties of the particulate organic molecules that are being oxidized. For instance, consider particles that consists of an internal mixture of molecules with relatively high $\kappa_{OA}$ (i.e., highly oxidized with higher O:C, and/or smaller molecules with lower MW) and molecules with relatively low $\kappa_{OA}$ (i.e., less oxidized with lower O:C, and/or larger molecules with higher MW), giving some average measured $\kappa_{OA}$. Upon heterogeneous oxidation, these two general classes of organic molecules may react differently. Several scenarios could lead to the oxidized particles being enriched in the lower $\kappa_{OA}$ molecules, which would decrease the average $\kappa_{OA}$ of the particles. First, due to their lower MW and expected high volatilities, the molecules with relatively high $\kappa_{OA}$ could preferentially evaporate from the particles after fragmentation compared with the lower $\kappa_{OA}$ molecules, leaving a larger fraction of low $\kappa_{OA}$ molecules in the





particle. The lower $\kappa_{OA}$ molecules could react to give less evaporation by either preferentially functionalizing instead of fragmenting or by starting with such low volatility that even the fragmentation products tend to have low enough volatility to remain in the particle phase. For these measurements during GoAmazon2014/5, this class of lower $\kappa_{OA}$ molecules could be represented by fresh or oxidized POA (see references above) or by BBOA

(e.g., Pósfai, 2004; Zhou et al., 2017). It is difficult to discern the exact reason for the decrease in $\kappa_{OA}$ at high O:C in this study due to the uncertain molecular composition of ambient OA, but these measurements warrant future OFR studies to investigate the effects of heterogeneous oxidation on the CCN properties of real ambient particles.

### 3.8    Estimating source contributions to potential SOA using multi-linear regression analysis

The results in Sects. 3.4–3.5 led to the conclusion that a dominant fraction of the SOA formation potential from oxidation of ambient air by OH, particularly during nighttime hours, was derived from gases that were not speciated or quantified during this campaign. Also, these gases could form SOA upon OH oxidation, but little or no SOA after $O_3$ oxidation, suggesting they tended not to contain C=C bonds. These conclusions are consistent with previous measurements of the oxidation of ambient air in an OFR in pine forest air in the US Rocky

Mountains (Palm et al., 2016, 2017) and in urban outflow downwind of Los Angeles (Ortega et al., 2016). In the analyses of the pine forest measurements, it was found that the unmeasured SOA-forming gases were likely to be S/IVOCs. Because the measured SOA formation correlated well with ambient MT, it was likely that the S/IVOCs were biogenic oxidation products (or were at least co-emitted with MT). With respect to ambient SOA-forming gases, the rural pine forest air system was relatively simple and was generally dominated by biogenic,

terpene-related gases.

A measurement of the total concentration of S/IVOCs during GoAmazon2014/5 was not available (as is typical for most field campaigns at present). However, information can still be extracted about the main sources contributing to the SOA formation potential from S/IVOCs present in ambient air by comparing with available VOC and/or tracer measurements. In this analysis, we make the assumption that the conclusion from the pine

forest measurements, specifically that VOCs and S/IVOCs from a given emission type correlate well with tracers from that same source, will also apply to all of the emission types at the T3 site.



The T3 site of GoAmazon2014/5 was chosen because it was expected (and was shown in Kourtchev et al. 2016) to be impacted by multiple distinct types of emissions. These include regional biogenic emissions (isoprene, MT, SQT, etc.), urban emissions from the city of Manaus, and local and regional biomass burning emissions. Unlike the previously mentioned results at the pine forest or the Los Angeles area, the maximum amount of SOA

formation in the OFR at T3 did not correlate well with any single SOA precursor gas, indicating the variable impacts of multiple sources. This conclusion can be drawn from the low correlations observed in the scatterplots of maximum SOA formation vs. precursors or tracers from each of the three emission types (MT, SQT [measured by SV-TAG], the sum of available biomass burning tracers, $NO_y$, NO, isoprene, acetonitrile, benzene, toluene, xylenes, and trimethylbenzenes) shown in Figs. S14 and S15 for wet and dry seasons,

respectively.

If the assumption holds that VOCs and S/IVOCs from a given emission type correlate with each other, then a multivariate relationship should exist, where the measured SOA formation should correlate well with the sum of measured concentration of VOCs/tracers of each source, multiplied by coefficients. The coefficients would quantify the relative contributions to potential SOA from VOCs + S/IVOCs from each source, relative to the

tracer. For this analysis, we used tracer gases that were likely to be dominated by a single type of source, including MT, SQT, and isoprene for biogenic emissions, NOy for urban emissions, and the sum of several measured BB tracers (vanillin, vanillic acid, syringol, and guaiacol) for biomass burning emissions. The background concentrations of the biogenic and BB tracers in air that did not contain emissions from those sources were near zero, and those tracers were all expected to react on roughly the same time scale on which

SOA formation occurred (on the order of a day or less). This makes these chosen tracers better suited for this type of analysis, since they were likely found only in the relatively fresh emissions that contain SOA forming gases, and were not measured in air after long range transport when the potential SOA would have already been formed. $NO_y$ is not itself an SOA-forming gas, but enhancements above the background were indicative of the total exposure of the air to urban sources, and it also accounted for dilution of the air in transport to the T3

site. For this analysis, a background of 0.7 ppb $NO_y$ was subtracted before performing the multilinear regression (MLR). Longer-lived tracers such as acetonitrile and benzene, were not suitable for this analysis, because their concentrations depended more on the long term history of the air. Also, gases such as benzene, toluene, and



xylene can be emitted from urban, biomass burning, and even biogenic sources, which makes them less distinct tracers of a given source type (e.g., Misztal et al., 2015).

Figure 12 illustrates the scatterplots of measured SOA formation vs. the amounts predicted by the MLR approach. The $R^2$ values increased substantially compared to the correlations with any individual precursors, up to 0.49 (0.30) for the wet (dry) season. Also shown are the diurnal profiles of estimated contributions to potential SOA from each of the three source types. This illustrates that the MLR approach can roughly match the diurnal profile of maximum SOA formation measured in the OFR by fitting coefficients to the diurnal profiles of measured tracers.

This analysis was carried out by allowing a single, fixed coefficient value for each tracer, i.e. implicitly assuming that the ratio of total SOA forming gases to the tracer was constant at all times of day and throughout each season. Given the natural variability of the atmosphere, this ratio is unlikely to be constant at all times (e.g., due to changing emission type compositions or degree of ambient photochemical aging). Ideally, the multilinear regression analysis could be performed as a function of time of day, which would allow the coefficient fits to vary with time of day. However, when performing the analysis this way, the correlation between independent variables rises to values sufficiently high that the multilinear fit can no longer distinguish between independent sources, and the analysis is no longer conclusive.

The average amounts and fractions of total SOA formation estimated from each of the biogenic, urban, and BB sources during each season are shown in Fig. 13a. Averages of 1.50 and 2.53 µg m$^{-3}$ were formed from ambient air during the wet and dry seasons for the times where data was available for SOA formation and all tracers. Of these amounts of potential SOA, 0.73 (48%), 0.67 (45%), and 0.10 (7%) µg m$^{-3}$ during the wet season and 1.76 (69%), 0.30 (12%), and 0.47 (18%) µg m$^{-3}$ during the dry season were attributed to biogenic, urban, and BB sources, respectively. These results indicate that biogenic SOA forming-gases were the most important contributors to measured potential SOA during both seasons. Urban sources contributed more than double the mass and nearly quadruple the fraction to potential SOA during the wet season compared to the dry season. BB sources of SOA-forming gases contributed almost five times more potential SOA mass during the dry season compared to the wet season. For reference, Fig. S16 shows these estimated contributions compared with the amount predicted from measured VOCs as in Sect. 3.5.





One way to help interpret these results is by comparing the average concentrations of the tracers in each season, along with the average potential SOA formation in the OFR, as shown in Fig. 13b. As expected (Martin et al., 2010), the BB tracers used in this analysis were found in much larger concentrations (~20x) during the dry season, which gives confidence in the larger mass contribution (x5) of BB-related gases to potential SOA. The

biogenic and NO$_y$ tracers were found in roughly equal concentrations in each season. This contrasts with the twice larger total contribution of urban SOA-forming gases during the wet season, vs. twice larger for biogenic sources during the dry season. Aromatic compounds were found in somewhat higher concentrations during the dry season, but those compounds also have a major biomass burning source, and in the dry season a larger proportion of these measured compounds was represented by benzene and toluene (representing less SOA

formation potential) compared to xylenes and trimethylbenzenes (representing more SOA formation potential). This suggests key differences between the average wet and dry season atmospheres. One hypothesis is that these differences could be related to changing ambient photochemistry between seasons. The 12-h average daytime solar irradiation during the wet season was 307 W m$^{-2}$, which was 23% less than the 398 W m$^{-2}$ during the dry season and suggests that photochemistry in ambient air was slower during the wet season. The

toluene:benzene ratio in ambient air at the T3 site was higher in the wet season (1.45) than the dry season (1.0). Since toluene reacts faster with OH radicals than benzene, a higher ratio in the wet season indicates "fresher" or less processed emissions arriving at T3 from the city of Manaus (de Gouw et al., 2005; Parrish et al., 2007). With slower ambient photochemistry, more urban SOA precursor gases could have survived the transport from Manaus leading to higher amounts of potential urban SOA formation in the OFR. In the dry

season, these gases may have already been oxidized in the atmosphere to form SOA en route to the T3 site, entering the OFR as OA and not contributing to potential SOA formation.

Stronger photochemistry could also explain the 2.4x larger biogenic potential SOA mass during the dry season. Measurements and models in Gu et al. (2017) showed that isoprene emissions were approximately 2x higher in the dry season than the wet season during this field campaign. We can make the assumption that other

biogenic gases (including MT and SQT) also exhibited higher emissions in the dry season. The stronger photochemistry could mean that there was a shift towards a higher relative ratio of biogenic S/IVOC concentrations to primary VOC/IVOC concentrations. Since the primary biogenic gas concentrations were very similar in both seasons (shown in Fig. 13b), the possibly higher biogenic S/IVOC concentrations in the dry season



could explain the larger potential SOA from that source. The very different spatial footprints of urban and biogenic emissions would then result in these different effects on potential SOA from each source at the T3 sites. These hypotheses should be tested with future modeling studies.

This analysis estimates the contributions from each of these three emission types to the concentrations of SOA forming gases (measured and unmeasured) at the T3 site. This provides information about what types of SOA could form upon further oxidation of this air at or downwind of the T3 site. Importantly, this analysis does not provide information about what amounts or fractions of the pre-existing (i.e. ambient) OA measured at the T3 site came from each of these sources. To investigate the sources of OA that impact the site and others in Amazonia, PMF analysis or other tracer analysis will be implemented in future work. However, it would seem plausible that the biogenic and biomass burning potential SOA sources observed here would also be important in formation of the OA on a regional scale, whereas the urban potential SOA source type may be more intense in the Manaus plume (within approximately the first day of transport) and less important on a regional scale.

The measurements at the T2 site were limited to a shorter period of time, and the available tracer measurements were less extensive, so multilinear analysis was not performed for the T2 site. Multilinear analysis was also not performed or needed for the SOA formation from $O_3$ oxidation at the T3 site, since Sect. 3.5 showed that all of the potential SOA formation can be roughly accounted for using the measured VOCs. In addition, the signal-to-noise of SOA formation from $O_3$ was low, which would limit this type of analysis.

## 4      Conclusions

During GoAmazon2014/5, ambient air was oxidized by OH or $O_3$ in an OFR in order to quantify (with high time resolution) the amount of potential SOA that could form from any precursors in ambient air. A range from 0 to as much as 10 µg m$^{-3}$ of potential SOA was formed in the OFR. This potential SOA formation was roughly a measure of the relative concentrations of SOA-forming gases (multiplied by their SOA yields) in the atmosphere, where the gases were measured by first converting them into more easily measurable particles. The potential to form SOA from ambient air changed with time of day, from one day to the next, and between the wet and dry seasons. As has been reported for previous field campaigns in a variety of locations, there were typically more SOA precursor gases found in ambient air during nighttime than during daytime. The amount of SOA from $O_3$ oxidation was consistent with the amount expected from the measured ambient precursors, but the amount



formed from OH oxidation was up to several times larger than could be accounted for with available measured gases. This provided further evidence that the unmeasured SOA-forming gases tended to not contain C=C bonds. These results suggest that during the day the high ambient OH is already converting most SOA precursors to SOA rapidly, while at night the lack of OH allows precursors to accumulate, especially those that

do not have C=C bonds and do not react with $O_3$ or $NO_3$. A multilinear regression analysis indicated that approximately two thirds of the potential SOA was biogenic in origin, while the remainder was mostly urban during the wet season and an equal mix of urban and biomass burning emissions during the dry season.

For the first time, SOA yields in the OFR were measured under ambient RH and temperature conditions, ambient external OHR levels, and using ambient aerosol as seeds for condensation. With consideration to many

factors that can affect the quantification of SOA yields in OFR experiments, the measurements presented herein increase the confidence of the conclusion that SOA yields in the OFR (particularly when performing measurements of the oxidation of ambient air) are similar to yields measured in large environmental chambers.

This work adds to the growing body of literature that employs an OFR to investigate SOA formation from ambient air. Such experiments are consistently suggesting that gases other than the commonly measured VOCs

are ubiquitous in the atmosphere, possibly having low volatilities and/or concentrations that make them difficult to measure, but with relatively high total potential to form SOA. In order to fully understand gas-to-particle SOA formation, we need to know more about these gases, including their identity, lifetime, reaction rates, SOA yields, deposition rates, etc., in order to be able to sufficiently model aerosol concentrations on regional and global scales.

**Data availability**

The data sets used in this publication are available at the ARM Climate Research Facility database for the GoAmazon2014/5 campaign (https://www.arm.gov/research/campaigns/amf2014goamazon).

**Competing interests**

The authors declare that they have no conflicts of interest.

**Acknowledgements**



Institutional support was provided by the Central Office of the Large Scale Biosphere Atmosphere Experiment in Amazonia (LBA), the National Institute of Amazonian Research (INPA), and Amazonas State University (UEA). We acknowledge support from the Atmospheric Radiation Measurement (ARM) Climate Research Facility, a user facility of the United States Department of Energy (DOE), Office of Science, sponsored by the Office of Biological

and Environmental Research, and support from the Atmospheric System Research (ASR) program of that office. Additional funding was provided by the Amazonas State Research Foundation (FAPEAM), the São Paulo State Research Foundation (FAPESP), the USA National Science Foundation (NSF), and the Brazilian Scientific Mobility Program (CsF/CAPES). The research was conducted under scientific license 001030/2012-4 of the Brazilian National Council for Scientific and Technological Development (CNPq). This research was supported by the U.S.

Department of Energy's (DOE) Atmospheric Science Program (Office of Science, BER, Grants No. DE-SC0016559 and DE-SC0011105), and the DOE SBIR program (DE-SC0011218), as well as NSF AGS-1360834 and EPA STAR 83587701-0. BBP is grateful for an EPA STAR graduate fellowship (FP-91761701-0). This manuscript has not been reviewed by EPA and no endorsement should be inferred. A portion of this research was performed using EMSL, a DOE Office of Science User Facility sponsored by the office of Biological and Environmental Research

and located at Pacific Northwest National Laboratory. We thank Dr. Francesco Canonaco for providing the SoFi software. G.I.VW. was supported by the NSF Graduate Research Fellowship (#DGE 1106400). SV-TAG data collection was made possible by NSF Atmospheric Chemistry Program 1332998, with instrument development supported by U.S. Department of Energy (DOE) SBIR grant DE-SC0004698.



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




**Figures:**

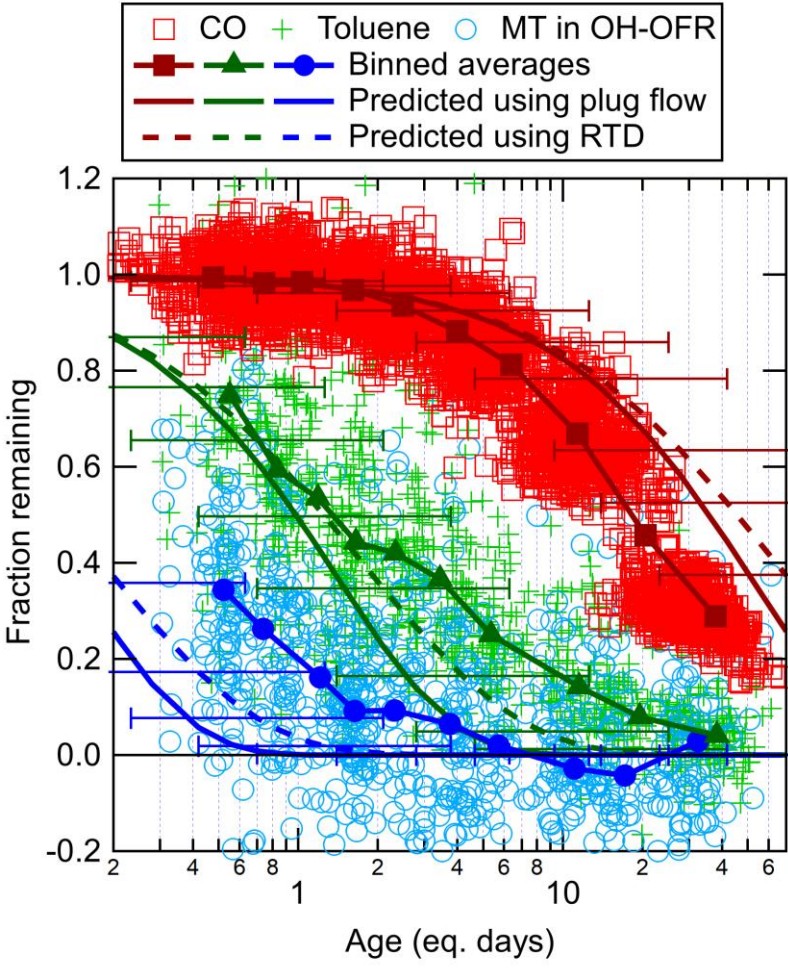

**Fig. 1.** Fraction of ambient toluene, ambient MT, and injected CO remaining after OH oxidation in the OFR, as a function of equation-estimated photochemical age (Peng et al., 2015). Binned averages of the fraction remaining are also shown, compared to the amount predicted to remain assuming either plug flow or using the residence time distribution (RTD) for particles from Lambe et al. (2011a). Factor-of-3 error bars are shown for the prediction using RTD, representing the uncertainty in the model-derived $OH_{exp}$ estimation equation (Li et al., 2015; Peng et al., 2015).



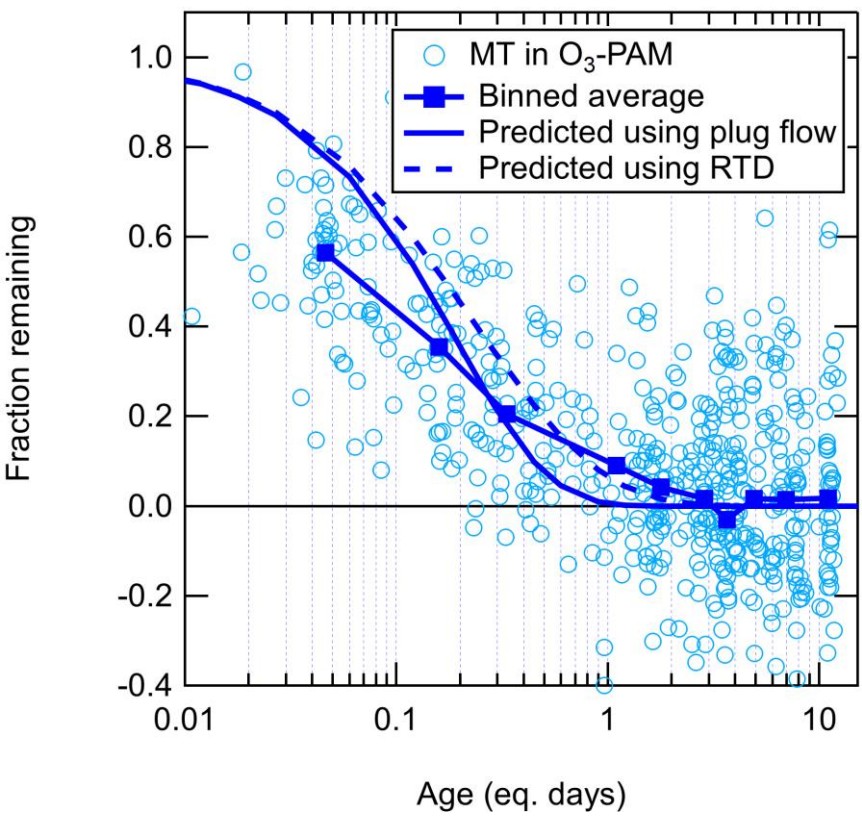

**Fig. 2.** Fraction of ambient MT remaining after $O_3$ oxidation in the OFR, as a function of photochemical age. Binned averages of the fraction remaining are also shown, compared to the amount predicted to remain assuming either plug flow or using the RTD of particles from Lambe et al. (2011a).





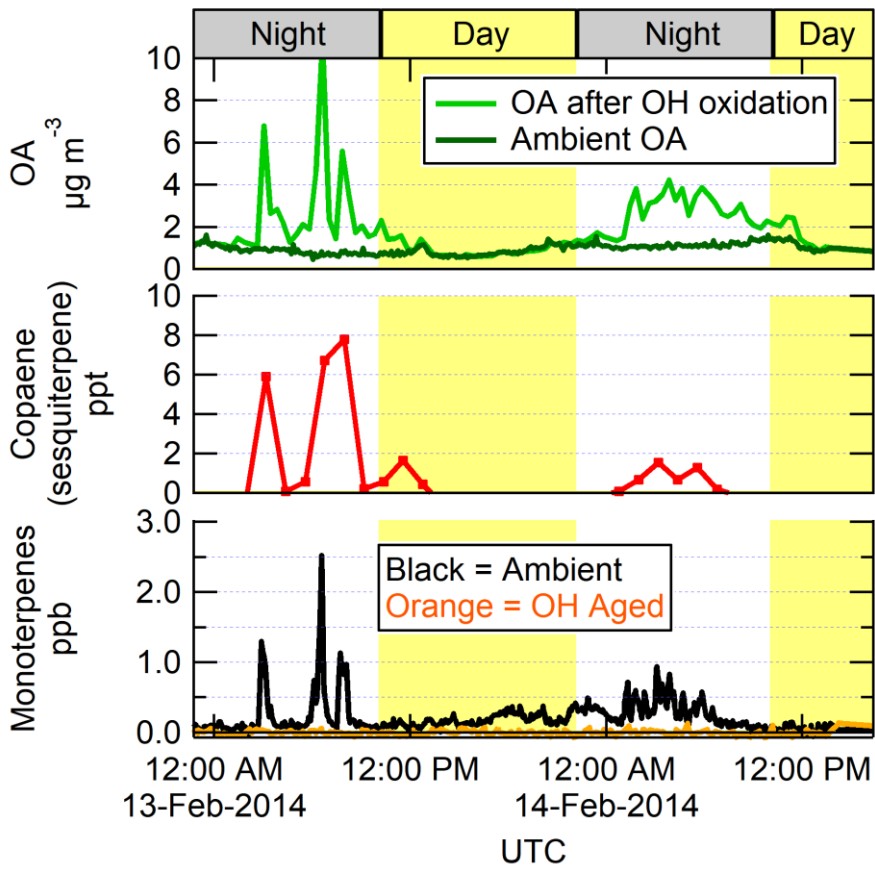

**Fig. 3.** An example of the time series of OA concentrations measured in ambient air and after OH oxidation of ambient air in the OFR at the T3 site, shown together with ambient copaene (a sesquiterpene, measured by SV-TAG) and monoterpenes (measured by PTR-TOF-MS before and after the OFR). Daytime hours are indicated with the yellow background. $OH_{exp}$ in the OFR was held constant throughout this time at approximately 3 eq. days. The SOA formed in the OFR is shown as measured, without the LVOC fate correction. In this example, the SOA formation from OH oxidation closely follows the availability of ambient biogenic gases, though the amount of SOA formed was substantially larger than could be formed from the measured ambient gases (see Sect. 3.5).





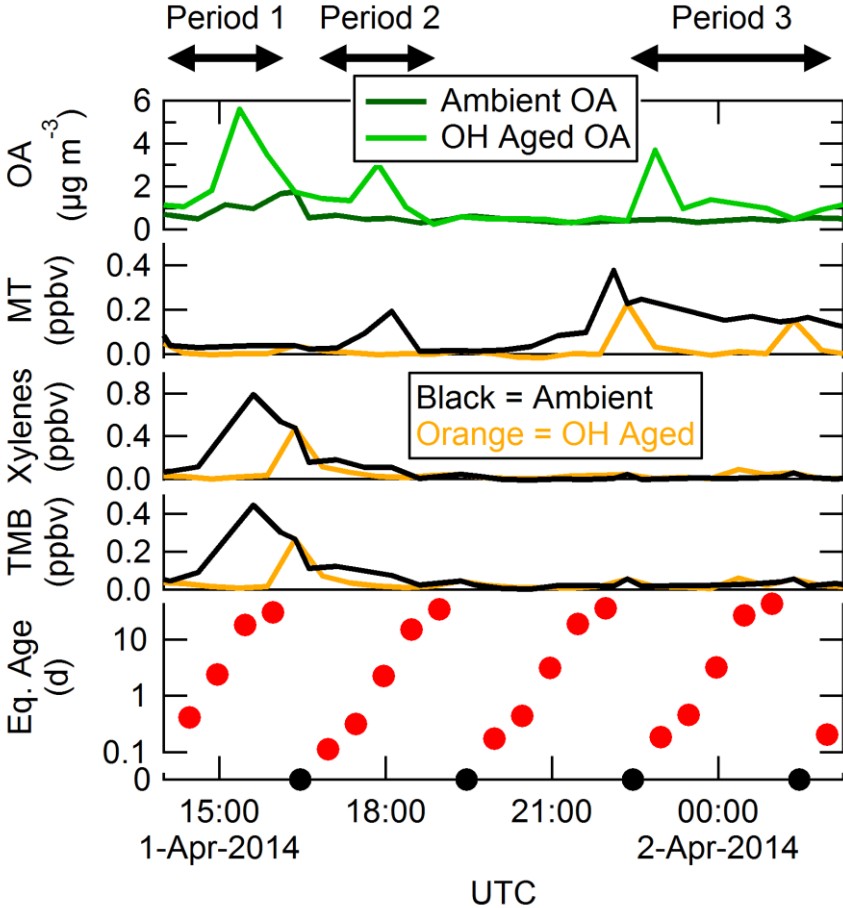

**Fig. 4.** An example of OA concentrations in ambient air and after OH oxidation of ambient air in the OFR at the T2 site, shown together with MT, xylenes, and trimethylbenzene (TMB) measured in ambient air and after OH oxidation. The $OH_{exp}$ is also shown (in eq. days). OH age was cycled through a range of exposures, including no added exposure (black circles) where none of the VOCs were reacted in the OFR. This example illustrates how SOA formation in the OFR can come from urban (Period 1), biogenic (Period 3), or mixed (Period 2) precursors, depending on ambient conditions.





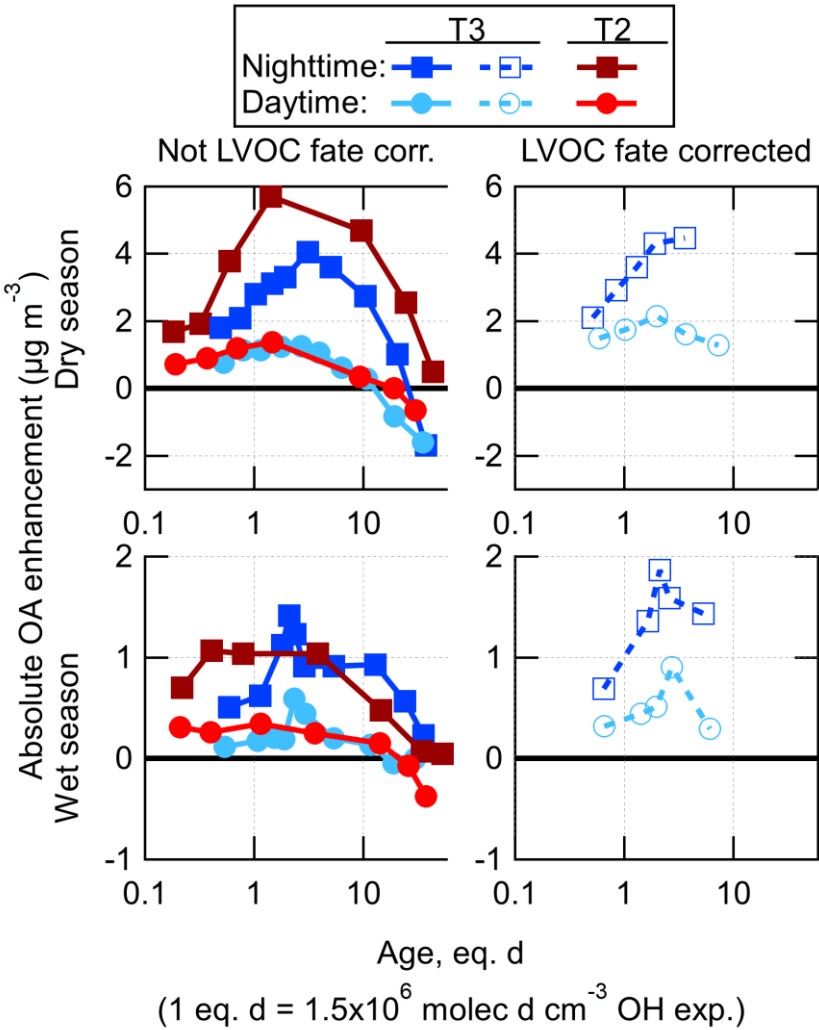

**Fig. 5.** Absolute OA enhancement after OH oxidation in the OFR as a function of photochemical age, shown as binned averages for the wet (bottom) and dry (top) seasons at both the T2 and T3 measurement sites, and split into daytime (06:00–18:00 LT) and nighttime (18:00–06:00 LT) data. This data is shown both not corrected (left) and corrected (right) for LVOC fate. Note that the scale of the y axis is different between the wet and dry season panels. The average ambient OA concentrations during the measurement times used here were 1.2 µg m⁻³ and 6.9 µg m⁻³ at T2 in the wet and dry seasons, and 1.3 µg m⁻³ and 9.5 µg m⁻³ at T3 in the wet and dry seasons, respectively.




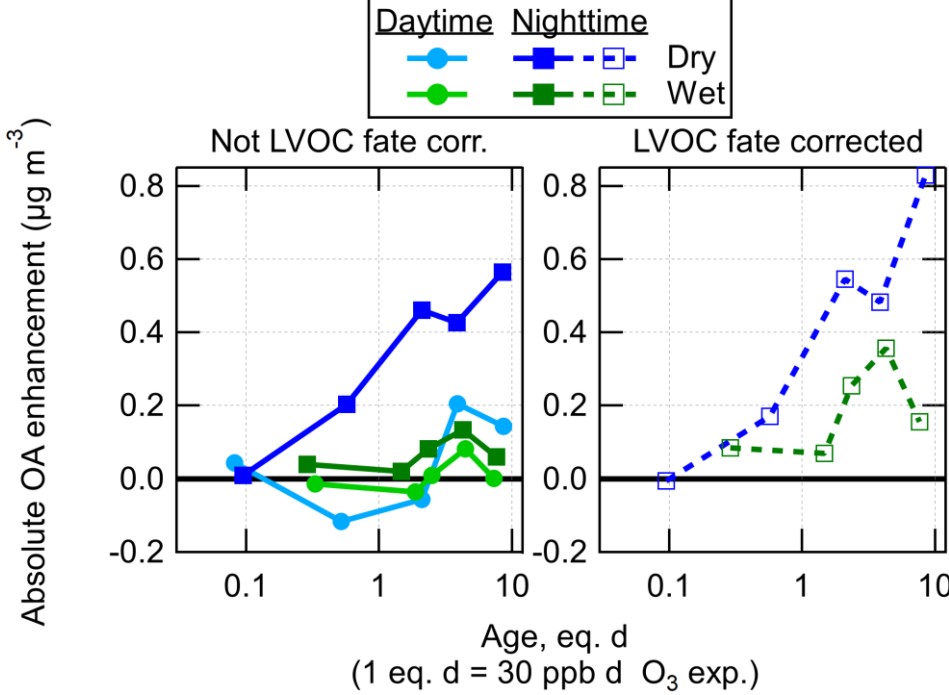

**Fig 6.** Absolute OA enhancement after $O_3$ oxidation in the OFR as a function of photochemical age, shown as binned averages for the wet and dry seasons at the T3 measurement site, and split into daytime (06:00–18:00 LT) and nighttime (18:00–06:00 LT) data. This data is shown both not corrected (left) and corrected (right) for LVOC fate.



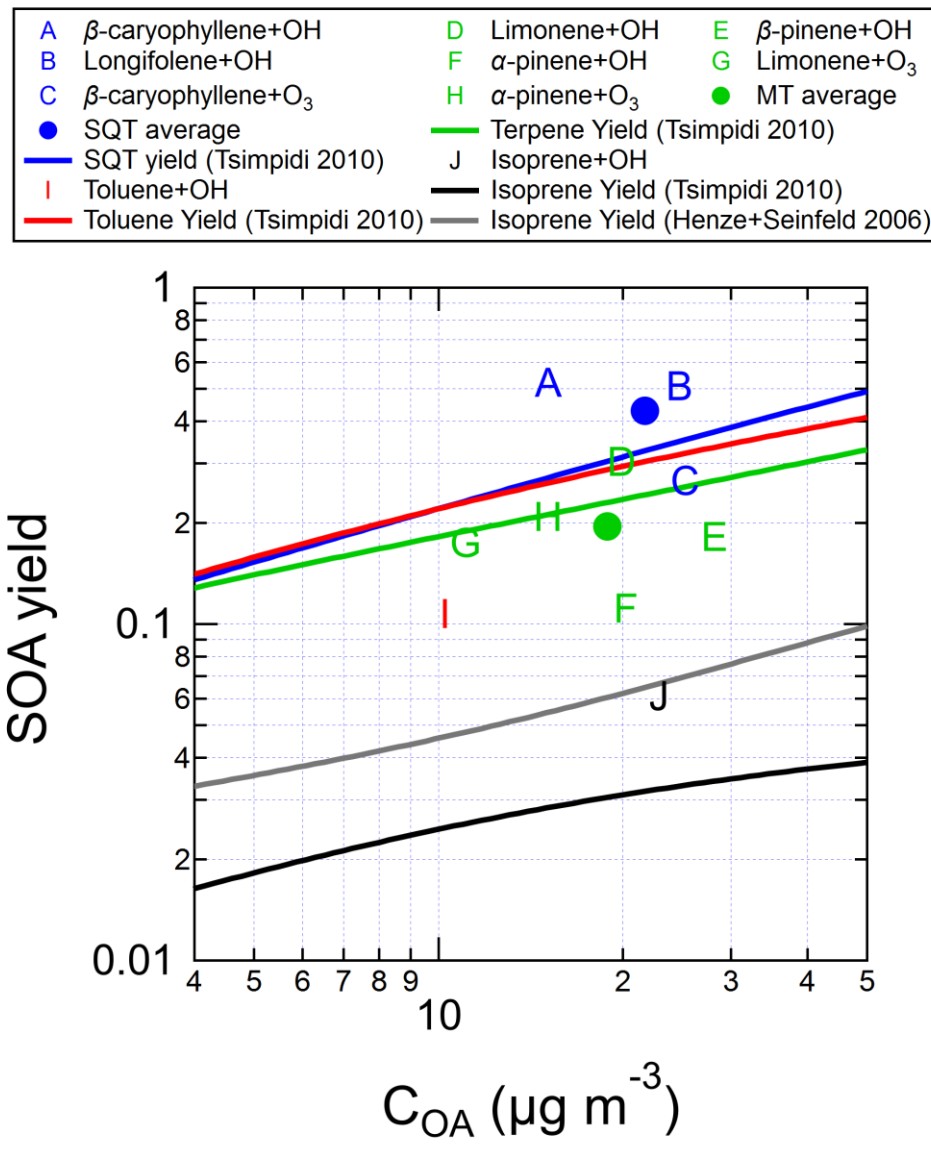

**Fig. 7.** SOA yields measured for individual VOCs in the OFR by standard addition into ambient air, as a function of OA concentration. Typical SOA yield parameterizations (derived from a chemical transport model, which was informed using environmental chamber experiments; Lane et al., 2008a, 2008b; Tsimpidi et al., 2010) are also shown. The VOCs were injected into ambient air at the entrance to the OFR, and aged between 0–5 eq. days. Data are corrected for LVOC fate.



**Fig. 8.** Measured SOA formation vs. the concentration of SOA predicted to form from the oxidation of ambient VOCs, shown for OH and O₃ oxidation during both wet and dry seasons. Regression lines and correlation coefficients are shown for each OFR type and season. Data are colored by local time of day. Measured SOA formation is corrected for LVOC fate.





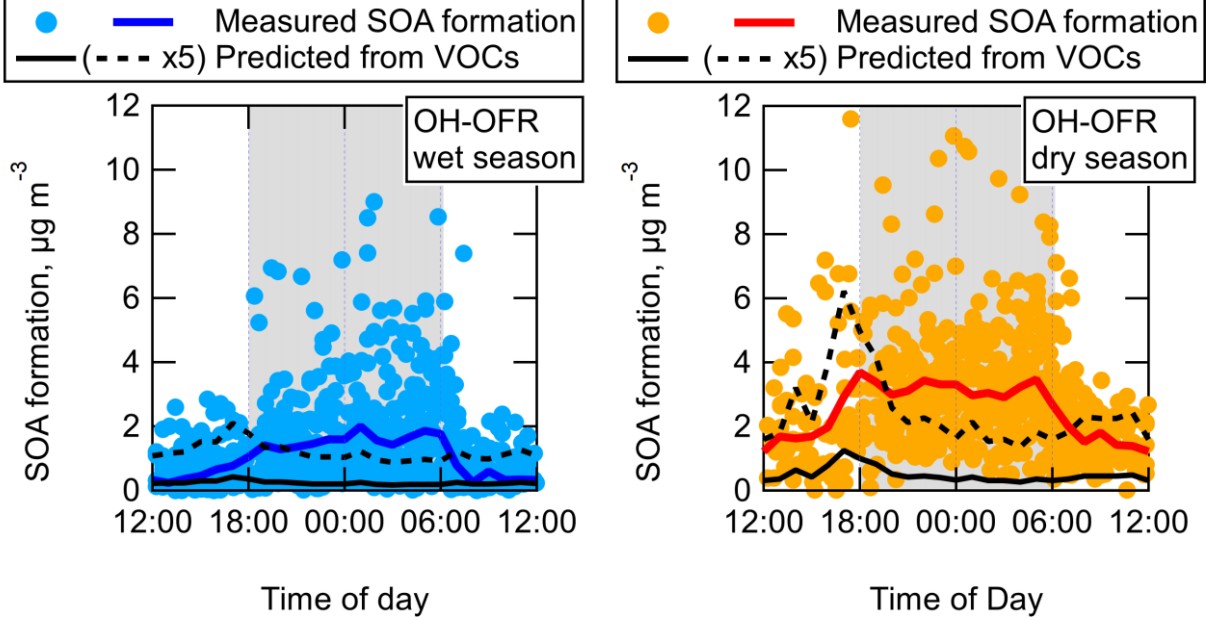

**Fig. 9.** Measured SOA formation vs. local time of day and mean diurnal cycles of measured and predicted SOA formation, shown for OH oxidation during both wet and dry seasons. For clarity, the predicted SOA from ambient VOCs is also shown multiplied by 5 for easier comparison of the trend relative to measured SOA formation.





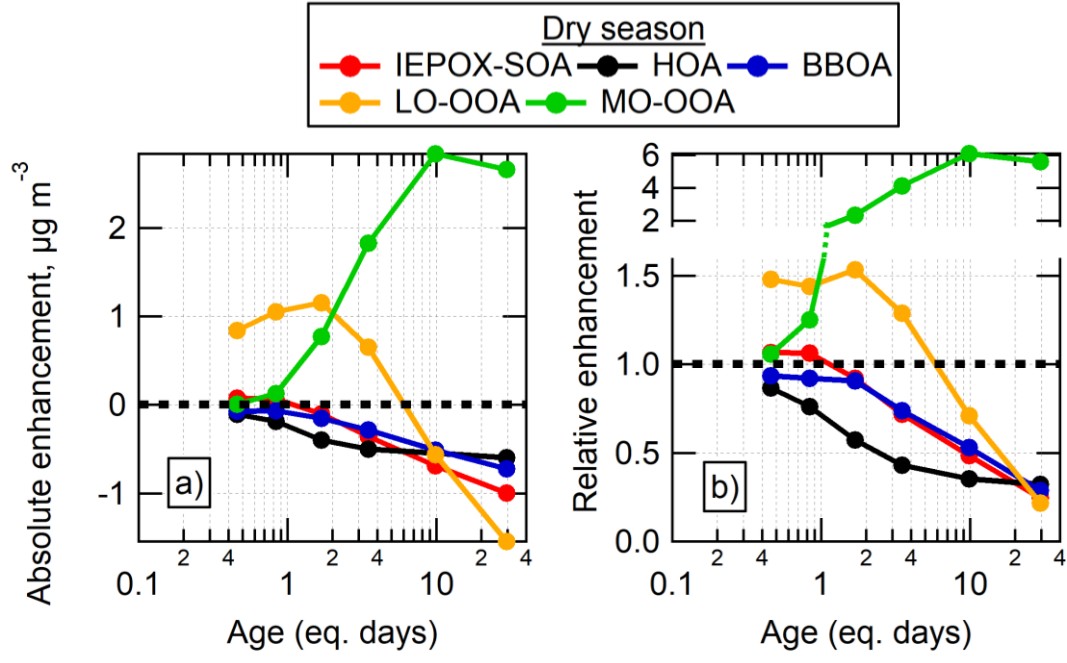

**Fig. 10.** Absolute (a) and relative (b) changes in PMF factors as a function of eq. days of OH aging in the OH-OFR for the dry season. Note that the y axis in panel (b) is split in order to more clearly show the region below a value of 1.





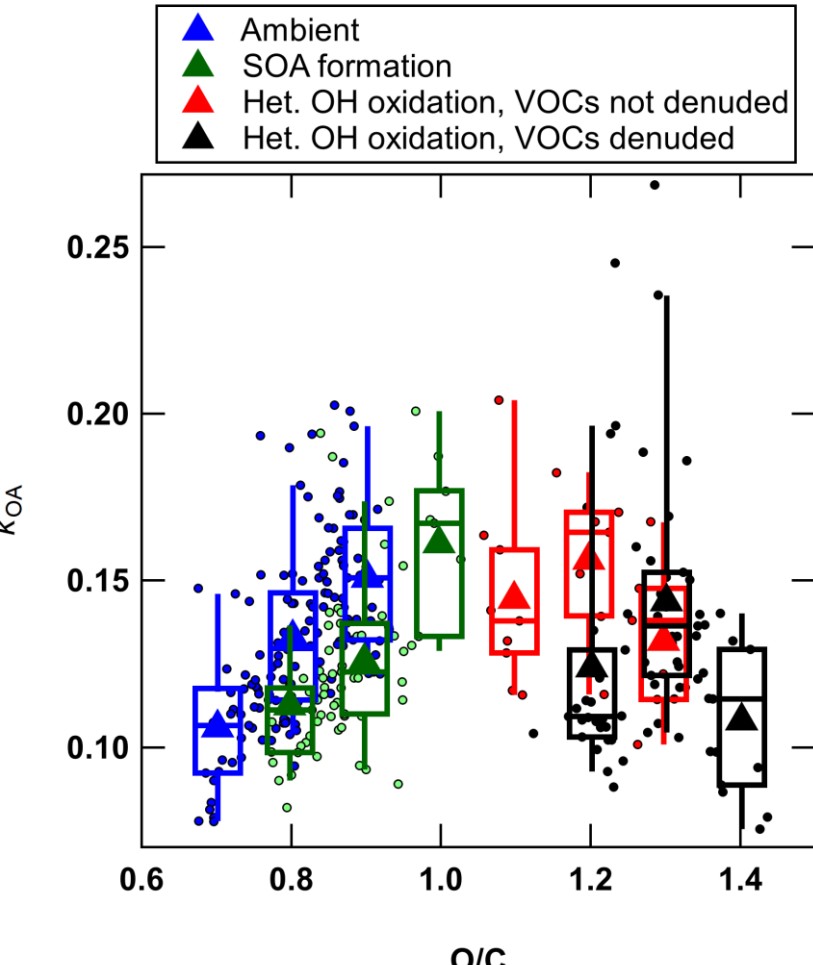

**Fig. 11.** Binned averages of hygroscopicity of OA ($\kappa_{OA}$) measured at 160 nm as a function of bulk O:C of the OA. The data includes ambient data, measurements after 1–3 eq. days OH aging to sample maximum SOA formation, and measurements after 12–44 eq. days aging sampled through (or not through) a gas denuder in order to sample the result of heterogeneous oxidation of pre-existing OA.





**Fig. 12.** Top: Maximum measured SOA enhancement from OH oxidation in the OFR at the T3 site during the wet and dry seasons, vs. the total amount predicted from multilinear regression analysis. Bottom: Diurnal average values of the maximum measured SOA formation from OH oxidation during each season, the amount attributed to each emission source, and the total amount predicted from all sources.





**Fig. 13.** b) The amounts and fraction of the total SOA formation from OH oxidation in the OFR at the T3 site that were attributed to biogenic, urban, and biomass burning emission types using multilinear regression analysis. a) Comparison of the average tracer concentrations and potential SOA formation during wet and dry seasons.