# Peer review of "Secondary organic aerosol formation from ambient air in an oxidation flow reactor in central Amazonia"

_Atmospheric Chemistry and Physics, 2017_

## Referee Comment (RC1) · Anonymous Referee #1 · 2 Oct 2017

This paper reports measurement of secondary organic aerosol (SOA) formation using an oxidation flow reactor (OFR) couples to an Aerosol Mass Spectrometer (AMS). The study has been conducted in two different sites of a very interesting area such as the central Amazonia. With this relatively new way of studying aerosol formation they were able to further oxidize ambient air using either OH or O3 and determine the potential SOA that can be formed starting from ambient VOC. The authors have shown that there are unmeasured gases, most probably not containing C=C, with relatively high SOA potential. Finally, they confirm that this approach (OFR + AMS) can be very useful in order to study SOA and to measure SOA yield at real ambient conditions.

I consider this study suitable for a publication in Atmospheric Chemistry and Physics. The paper is very well written and all the results are presented very clearly. However, I have a non-scientific consideration for the authors. I would personally prefer to see such study in a shorter form and move all the technical details and extra discussion in the supplementary information. I have the feeling that the long discussion remove emphasis from the final scientific message. On the other end, I am aware of the fact that ACP accepts long and detailed studies. Therefore, it is up to the authors to decide if they want to keep it as long as it is now or they prefer to shorten the main text focalizing on the main results. Beside that I would accept the manuscript as it is beside very few minor comments.

Page 3 from line 9 to line 24: Although not mandatory, when discussing about the volatility of the organic compounds (SVOCs and IVOCs), I would add extra references mentioning the Highly Oxygenated Molecules and the ELVOC. I think that this would be a nice small piece of extra information.

Fig.3: In the monoterpene panel the authors mention the ambient measurements (black) and OH aged (orange). Although I understand what they mean, it is a bit misleading to have aged monoterpene. To avoid confusion, I would use a different term such as non-reacted monoterpene or anything else that doesn't lead to any confusion.

Fig.4: Same as figure 3.

---

## Referee Comment (RC2) · Anonymous Referee #2 · 9 Nov 2017

Review of

**Secondary organic aerosol formation from ambient air in an oxidation flow reactor in central Amazonia**

By Palm et al.

General comments

Palm et al. investigated the formation of secondary organic aerosol (SOA) from ambient air at two sites downwind of Manaus, Brazil. The experiments were conducted using Potential Aerosol Mass (PAM) oxidation flow reactor (OFR) and were observed by an array of particle and gas measurements. In addition, the study compared the SOA formed in the PAM reactor with predicted SOA from the measured ambient SOA precursors and their yields. The study found that the SOA enhancement could come from unmeasured semi-volatile and intermediate volatility gases. The sources of unmeasured SOA precursors were suggested to be biogenic, urban, and biomass burning emissions. I think the study contributed in exploring PAM OFR capacity to study atmospheric oxidation process in a field campaign. There are some parts that need clarifications and few typing errors. However, the manuscript overall reads well. I recommend accepting the manuscript for publication in the Atmospheric Chemistry and Physics journal after revisions.

Specific comments

1. Pg. 6 Ln. 11: It is suggested that variation of the estimated $OH_{exposure}$ could not be different more than a factor of 2 if the true OH reactivity (OHR) at T3 site was different from values at T0a site used as an assumption. Where is this factor of 2 coming from? Has sensitivity test been done? Please provide some information.

2. Pg. 7 Ln. 9: The reagent ion for PTR-TOF-MS was changed between IOP1 and IOP2 measurement periods. Were there specific reason(s) for changing the reagent ion? Also, would the measurement results be comparable between the two periods?

3. Pg. 15 Ln. 1: How was the wind direction between T2 site and Manaus? If the wind was not blowing to T2 from Manaus, the closer proximity of the site would not lead to increase in the urban emissions at T2. It may be good to add information of wind direction/trajectory in Section 2.1 (see technical comment #2).

4. Pg. 18 Lns. 12-14, Figure 8: The $R^2$ of measured vs. predicted SOA formation are low for all cases except the OH-OFR wet season. I am not convinced with the prediction approach. The typical chamber yield seems that it is not able to entirely capture SOA formation in the ambient of this study. This could be due to a more complex mixture of S/IVOCs in the atmosphere compared to in the chamber experiments. Thus, I would be more careful in interpreting the linear regressions results (i.e., slope), as the datasets (measured and predicted) do not show associations.

5. Pg. 20 Lns. 20-23: I think it is necessary to provide more information regarding PMF analysis using ME-2 algorithm (SoFi). In the earlier paragraphs (Pg. 20 Lns. 7-8), it is said that interpretation of factors from ambient OA is provided elsewhere. Hence, the details and

interpretation of factors from OFR OA should be provided here. The additional information such as:

   a. How the constrain was applied into the analysis (e.g., a-value or anchor) and how the solution was selected.

   b. Evaluation the unconstrained factors solution before constraining the factors solution. Previous studies recommended examining the unconstrained factors to determine constraint(s) required for improving the factors solution (Crippa et al., 2014; Fröhlich et al., 2015).

6. Pg. 21 Lns. 7-9, Figure 10: The plateau of LO-OOA looks more of a slight increase because it was followed by a significant decrease. The LO-OOA was gradually increasing from around 0.5 eq. day, reaching a peak at around 2 eq. day, and then continuously decreasing.

7. Pg. 21 Lns. 16-17: It is unclear what the SOA from PMF analysis results is. It is because PMF analysis yielded LO-OOA and MO-OOA which can be referred as the SOA factor. In case of the dry season, IEPOX-SOA is also an SOA factor.

8. Pg. 23 Ln. 3: The authors refer to Thalman et al. (2017) for details analysis of ambient OA hygroscopicity ($K_{org}$). Thalman et al. (2017) reported $K_{org}$ for particles with diameter range between ~90 to ~180 nm and showed that $K_{org}$ was independent of particle size. Is there any reason for selecting $K_{org}$ of 160 nm mobility diameter particles?

9. Pg. 26 Lns. 11-14 and Pg. 27 Lns. 3-4: Here, the authors mention about multivariate relationship or multilinear regression (MLR). I think it would be good to provide the MLR model and coefficient values. Hence, it is clear how the SOA formation was predicted (Figure 12).

Technical comments

1. Pg. 4 Ln. 24: What does GoAmazon2014/15 stand for? Have the description of field campaign here rather than on Pg. 5 Ln. 5.

2. Pg. 5 Ln. 12: Add information that T2 site is downwind of Manaus. It is unclear here whether it is downwind or upwind.

3. Pg. 6 Ln. 24: What is reference(s) for the typical 24 average ambient O3 concentration? Or is it from measurement at the site? Please clarify.

4. Pg. 7 Ln. 12: What does SQT stand for?

5. Pg. 8 Ln. 12: Does it mean the ambient OA concentration is an average of measurements before and after OFR sampling? Please clarify.

6. Pg. 10 Ln. 10: What does MT stand for?

7. Pg. 10 Ln. 17: The compounds and their acronyms should be provided earlier in the text. See technical comments #4 and 6.

8. Pg. 14 Ln. 7: What does LT mean?

9. Pg. 17 Lns. 19-22: Which large chamber studies that OA mass concentrations are a factor of 2 comparable with the present study? Please add references. It may also be good to add the OA mass from previous chamber studies in Table S1 for comparison.

10. Pg. 18 Ln. 11: Remove double from C=C bonds.

11. Pg. 20 Lns. 9-13:

Add reference for IEPOX-SOA: Budisulistiorini et al. (2013)

Add reference for Fac91: Robinson et al. (2011), Budisulistiorini et al. (2015)

12. Pg. 22 Lns. 21-25: Add reference(s) for observation of SOA-forming gases at the site (or nearby locations) during daytime and nighttime.

13. Figures 3 and 4: The unit of x axis (time) is better in Local Time (LT) because in the text you use LT.

14. Figure 13: Are the labels (a) and (b) on figure caption switched? Please check.

References

Budisulistiorini, S.H., Canagaratna, M.R., Croteau, P.L., Marth, W.J., Baumann, K., Edgerton, E.S., Shaw, S.L., Knipping, E.M., Worsnop, D.R., Jayne, J.T., Gold, A., Surratt, J.D., 2013. Real-Time Continuous Characterization of Secondary Organic Aerosol Derived from Isoprene Epoxydiols in Downtown Atlanta, Georgia, Using the Aerodyne Aerosol Chemical Speciation Monitor. Environ. Sci. Technol. 47, 5686–5694. doi:10.1021/es400023n

Budisulistiorini, S.H., Li, X., Bairai, S.T., Renfro, J., Liu, Y., Liu, Y.J., McKinney, K.A., Martin, S.T., McNeill, V.F., Pye, H.O.T., Nenes, A., Neff, M.E., Stone, E.A., Mueller, S., Knote, C., Shaw, S.L., Zhang, Z., Gold, A., Surratt, J.D., 2015. Examining the effects of anthropogenic emissions on isoprene-derived secondary organic aerosol formation during the 2013 Southern Oxidant and Aerosol Study (SOAS) at the Look Rock, Tennessee, ground site. Atmos. Chem. Phys. Discuss. 15, 7365–7417. doi:10.5194/acpd-15-7365-2015

Crippa, M., Canonaco, F., Lanz, V.A., Äijälä, M., Allan, J.D., Carbone, S., Capes, G., Ceburnis, D., Dall'Osto, M., Day, D.A., DeCarlo, P.F., Ehn, M., Eriksson, A., Freney, E., Hildebrandt Ruiz, L., Hillamo, R., Jimenez, J.L., Junninen, H., Kiendler-Scharr, A., Kortelainen, A.-M., Kulmala, M., Laaksonen, A., Mensah, A.A., Mohr, C., Nemitz, E., O'Dowd, C., Ovadnevaite, J., Pandis, S.N., Petäjä, T., Poulain, L., Saarikoski, S., Sellegri, K., Swietlicki, E., Tiitta, P., Worsnop, D.R., Baltensperger, U., Prévôt, A.S.H., 2014. Organic aerosol components derived from 25 AMS data sets across Europe using a consistent ME-2 based source apportionment approach. Atmos. Chem. Phys. 14, 6159–6176. doi:10.5194/acp-14-6159-2014

Fröhlich, R., Crenn, V., Setyan, A., Belis, C.A., Canonaco, F., Favez, O., Riffault, V., Slowik, J.G., Aas, W., Aijälä, M., Alastuey, A., Artiñano, B., Bonnaire, N., Bozzetti, C., Bressi, M., Carbone, C., Coz, E., Croteau, P.L., Cubison, M.J., Esser-Gietl, J.K., Green, D.C., Gros, V., Heikkinen, L., Herrmann, H., Jayne, J.T., Lunder, C.R., Minguillón, M.C., Močnik, G., O'Dowd, C.D., Ovadnevaite, J., Petralia, E., Poulain, L., Priestman, M., Ripoll, A., Sarda-Estève, R., Wiedensohler, A., Baltensperger, U., Sciare, J., Prévôt, A.S.H., 2015. ACTRIS ACSM intercomparison – Part 2: Intercomparison of ME-2 organic source apportionment results from 15 individual, co-located aerosol mass spectrometers. Atmos. Meas. Tech. 8, 2555–2576. doi:10.5194/amt-8-2555-2015

Robinson, N.H., Newton, H.M., Allan, J.D., Irwin, M., Hamilton, J.F., Flynn, M., Bower, K.N., Williams, P.I., Mills, G., Reeves, C.E., McFiggans, G., Coe, H., 2011. Source attribution of Bornean air masses by back trajectory analysis during the OP3 project. Atmos. Chem. Phys. 11, 9605–9630. doi:10.5194/acp-11-9605-2011

Thalman, R., de Sá, S.S., Palm, B.B., Barbosa, H.M.J., Pöhlker, M.L., Alexander, M.L., Brito, J., Carbone, S., Castillo, P., Day, D.A., Kuang, C., Manzi, A., Ng, N.L., Sedlacek III, A.J., Souza, R., Springston, S., Watson, T., Pöhlker, C., Pöschl, U., Andreae, M.O., Artaxo, P., Jimenez, J.L., Martin, S.T., Wang, J., 2017. CCN activity and organic hygroscopicity of aerosols downwind of an urban region in central Amazonia: seasonal and diel variations and impact of

anthropogenic emissions. Atmos. Chem. Phys. 17, 11779–11801. doi:10.5194/acp-17-11779-2017

---

## Author Comment (AC1) · 6 Dec 2017

**Response to reviewers for "Secondary organic aerosol formation from ambient air in an oxidation flow reactor in central Amazonia"**

Brett B. Palm, Suzane S. de Sá, Douglas A. Day, Pedro Campuzano-Jost, Weiwei Hu, Roger Seco, Steven J. Sjostedt, Jeong-Hoo Park, Alex B. Guenther, Saewung Kim, Joel Brito, Florian Wurm, Paulo Artaxo, Ryan Thalman, Jian Wang, Lindsay D. Yee, Rebecca Wernis, Gabriel Isaacman-VanWertz, Allen H. Goldstein, Yingjun Liu, Stephen R. Springston, Rodrigo Souza, Matt K. Newburn, M. Lizabeth Alexander, Scot T. Martin, and Jose L. Jimenez

We thank the reviewers for their comments on our paper. To facilitate the review process, we have copied the reviewer comments in black text. Our responses are in regular blue font. We have responded to all the referee comments and made alterations to our paper (**in bold text**).

**Anonymous Referee #1**

R1.0. This paper reports measurement of secondary organic aerosol (SOA) formation using an oxidation flow reactor (OFR) couples to an Aerosol Mass Spectrometer (AMS). The study has been conducted in two different sites of a very interesting area such as the central Amazonia. With this relatively new way of studying aerosol formation they were able to further oxidize ambient air using either OH or O3 and determine the potential SOA that can be formed starting from ambient VOC. The authors have shown that there are unmeasured gases, most probably not containing C=C, with relatively high SOA potential. Finally, they confirm that this approach (OFR + AMS) can be very useful in order to study SOA and to measure SOA yield at real ambient conditions. I consider this study suitable for a publication in Atmospheric Chemistry and Physics. The paper is very well written and all the results are presented very clearly.

R1.1. However, I have a non-scientific consideration for the authors. I would personally prefer to see such study in a shorter form and move all the technical details and extra discussion in the supplementary information. I have the feeling that the long discussion remove emphasis from the final scientific message. On the other end, I am aware of the fact that ACP accepts long and detailed studies. Therefore, it is up to the authors to decide if they want to keep it as long as it is now or they prefer to shorten the main text focalizing on the main results. Beside that I would accept the manuscript as it is beside very few minor comments.

We thank the reviewer for her/his thoughtful comments regarding the paper format. There are pros and cons to both types of manuscripts. This is one of the first studies of its type, and thus documenting key technical details in the main paper is important for the credibility of the work in the community. We note that we already tried to keep less-important technical aspects in the Supp. info, which is already extensive (21 pages with 50 figure panels). Thus we prefer to keep the current format for this work.

R1.2. Page 3 from line 9 to line 24: Although not mandatory, when discussing about the volatility of the organic compounds (SVOCs and IVOCs), I would add extra references mentioning the Highly Oxygenated Molecules and the ELVOC. I think that this would be a nice small piece of extra information.

We have modified this text at Pg. 3 Ln. 12 as follows to address this point:

"The most volatile organics are called volatile organic compounds (VOCs) and are found almost exclusively in the gas phase, while the lowest volatility compounds **(e.g., extremely low volatility compounds (ELVOCs), Ehn et al., 2014)** are found almost entirely in the particle phase as OA."

R1.3. Fig.3: In the monoterpene panel the authors mention the ambient measurements (black) and OH aged (orange). Although I understand what they mean, it is a bit misleading to have aged monoterpene. To avoid confusion, I would use a different term such as non-reacted monoterpene or anything else that doesn't lead to any confusion.

We have changed the legends in Fig. 3 to the following to avoid this problem. Note that Fig. 3 already includes the daylight period as a yellow background. We have added a grey background to the nighttime periods for clarity. The second sentence of the Fig. 3 caption now reads: "Daytime **(nighttime)** hours are indicated with the yellow **(grey)** background."

[Figure]

R1.4. Fig.4: Same as figure 3.

We have changed the legends in Fig. 4 to the following to avoid this problem. We have also added daytime (yellow) and nighttime (grey) background colors as in Fig. 3, and added the following sentence to the Fig. 4 caption: **"Daytime (nighttime) hours are indicated with the yellow (grey) background."**

[Figure]

**Anonymous Referee #2**

R2.0. Palm et al. investigated the formation of secondary organic aerosol (SOA) from ambient air at two sites downwind of Manaus, Brazil. The experiments were conducted using Potential Aerosol Mass (PAM) oxidation flow reactor (OFR) and were observed by an array of particle and gas measurements. In addition, the study compared the SOA formed in the PAM reactor with predicted SOA from the measured ambient SOA precursors and their yields. The study found that the SOA enhancement could come from unmeasured semi-volatile and intermediate volatility gases. The sources of unmeasured SOA precursors were suggested to be biogenic, urban, and biomass burning emissions. I think the study contributed in exploring PAM OFR capacity to study atmospheric oxidation process in a field campaign. There are some parts that need clarifications and few typing errors. However, the manuscript overall reads well. I recommend accepting the manuscript for publication in the Atmospheric Chemistry and Physics journal after revisions.

We thank the reviewer for these thoughtful comments.

Specific comments

R2.1. Pg. 6 Ln. 11: It is suggested that variation of the estimated OH exposure could not be different more than a factor of 2 if the true OH reactivity (OHR) at T3 site was different from values at T0a site used as an assumption. Where is this factor of 2 coming from? Has sensitivity test been done? Please provide some information.

To clarify this statement, we have changed the sentence starting at Pg. 6 Ln. 11 to read:

**"While the true OHR at any given time** at the site was **likely** different from the average in Williams et al. (2016) **due to natural variability or other reasons, empirical estimates suggest that** the model-estimated $OH_{exp}$ could be different by no more than a factor of 2 **over the range of reasonable ambient OHR values."**

R2.2. Pg. 7 Ln. 9: The reagent ion for PTR-TOF-MS was changed between IOP1 and IOP2 measurement periods. Were there specific reason(s) for changing the reagent ion? Also, would the measurement results be comparable between the two periods?

We have changed the text at Pg. 7 Ln. 9 to the following to address this question:

"Ambient and OFR-oxidized VOC concentrations were sampled during the entire campaign using an IONICON proton-transfer-reaction time-of-flight mass spectrometer (PTR-TOF-MS; Jordan et al., 2009a, 2009b; Müller et al., 2013)**. For scientific reasons external to the scope of this manuscript, this instrument** sampled using $H_3O^+$ as the reagent ion during IOP1 and $NO^+$ as the reagent ion during IOP2. **Sensitivities were calibrated independently for each reagent ion in order to maintain quantification across seasons."**

R2.3. Pg. 15 Ln. 1: How was the wind direction between T2 site and Manaus? If the wind was not blowing to T2 from Manaus, the closer proximity of the site would not lead to increase in the urban

emissions at T2. It may be good to add information of wind direction/trajectory in Section 2.1 (see technical comment #2).

For clarification, we changed the text at Pg. 5 Ln 12 from "located approximately 10 km west of Manaus" to read:

**"located approximately 10 km west (downwind) of Manaus"**

At Pg. 5 Ln. 14, we have added:

**"Prevailing wind direction is illustrated by back trajectories shown in Fig. 3 of Martin et al. (2016)."**

R2.4. Pg. 18 Lns. 12-14, Figure 8: The R2 of measured vs. predicted SOA formation are low for all cases except the OH-OFR wet season. I am not convinced with the prediction approach. The typical chamber yield seems that it is not able to entirely capture SOA formation in the ambient of this study. This could be due to a more complex mixture of S/IVOCs in the atmosphere compared to in the chamber experiments. Thus, I would be more careful in interpreting the linear regressions results (i.e., slope), as the datasets (measured and predicted) do not show associations.

The point that is being made at this point in the paper is that the amount of SOA formed in the OFR cannot be explained by the measured VOC precursors. Later in the paper (Fig. 12) we try to account for S/IVOC precursors, which leads to better correlation. The point about the VOCs can be made with the average of the data, leading to the same conclusion. For example for the OH-OFR in the dry season, the average predicted SOA is 2.7 $\mu g\ m^{-3}$ while the average predicted amount is 0.47 $\mu g\ m^{-3}$. The ratio of these numbers is 5.7, which is similar to the regression slope in that case (6.5), leading to the same conclusion: the measured VOCs are insufficient to explain the amount of SOA formed, and other non-measured species must make a major contribution to SOA formation from OH oxidation. To make these clearer, we have added the average values from each panel as an additional datapoint (larger black diamond) on top of the individual datapoints, as shown below:

[Figure]

"Fig. 8. Measured SOA formation vs. the concentration of SOA predicted to form from the oxidation of ambient VOCs, shown for OH and $O_3$ oxidation during both wet and dry seasons. **Regression lines, correlation coefficients, and average measured and predicted SOA are shown for each OFR type and season. Standard errors of the mean of all average measured and predicted SOA values were smaller than the size of the marker, and are thus not shown.** Data are colored by local time of day. Measured SOA formation is corrected for LVOC fate."

The issue of the yields has just been addressed at this point in the paper, with section 3.4 and Fig. 7, which show that the yields in the OFR *under the conditions of GoAmazon* are close to those used in the model, and thus differences in the yield cannot explain the large underestimations observed in Fig. 8.

Since the key point being made with Figure 8 and the text under discussion here concerns the amount of SOA predicted vs. measured across the campaign, the correlation coefficients are secondary to this analysis. It is understood that they will be low if key precursors (S/IVOC) are missing. And in addition there is a substantial level of noise in the individual 3-min-average data points shown in Figure 8, which is especially important for the low SOA formation amounts in the $O_3$-OFR cases.

Because the complexity of ambient S/IVOCs cannot be captured by correlations with VOCs, the approach in Sect. 3.5 is insufficient to fully explain the results (though Sect. 3.5 does serve to clearly illustrate the overall pattern of the measurements). It is for this reason that we included the more detailed analysis in Sect. 3.8. To address this comment, we have made the following changes to the text:

At Pg 18 Ln. 14, we added the words **"for reference"** to the end of the sentence.

At Pg. 18 Ln. 17, the text now reads:

"OH oxidation of ambient air produced on average 6.5–8 times more SOA than could be accounted for from ambient VOCs**, which is consistently observed in either the averaged data or the linear regressions**. This is consistent with previous OFR measurements, suggesting that typically unmeasured ambient **S/IVOC** gases play a substantial role in ambient SOA formation from OH oxidation **(see Sect. 3.8 for more analysis to explore this concept)."**

R2.5. Pg. 20 Lns. 20-23: I think it is necessary to provide more information regarding PMF analysis using ME-2 algorithm (SoFi). In the earlier paragraphs (Pg. 20 Lns. 7-8), it is said that interpretation of factors from ambient OA is provided elsewhere. Hence, the details and interpretation of factors from OFR OA should be provided here. The additional information such as: a. How the constrain was applied into the analysis (e.g., a-value or anchor) and how the solution was selected. b. Evaluation the unconstrained factors solution before constraining the factors solution. Previous studies recommended examining the unconstrained factors to determine constraint(s) required for improving the factors solution (Crippa et al., 2014; Fröhlich et al., 2015).

As noted on Pg. 20 Ln. 4, this analysis is only meant to provide information about how factors with characteristic mass spectral features (e.g., BBOA with elevated relative signal at *m/z* 60) change upon oxidation in an OFR. We did not intend to fully interpret these factors in the ambient air that entered the OFR, as documenting all the relevant details requires a separate manuscript in preparation (by a different first author). However, given that these factors are similar to those found previously in the Amazon (Chen et al., 2015) and in dozens of other studies, we do not believe that a very detailed description is necessary for drawing the conclusions presented herein. Instead we can mostly rely on literature references, since the results are similar to those published previously for the Amazon and elsewhere.

To clarify the details that are important in this analysis and the similarity to past results, we have changed the text at Pg. 20 Ln. 1-2 as:

"First, PMF was applied to only the unoxidized measurements through the OFR. The resulting PMF factors were similar to the factors identified in ambient air (de Sá et al., 2017), **and also similar to those observed previously at a nearby site in the Amazon during the AMAZE-2008 campaign (Chen et al., 2015)."**

And we have also changed the text on Pg. 20 Ln. 20 to read:

"For the wet season, PMF of the OH-aged aerosol was performed with a total of 6 factors, using the Source Finder analysis software (SoFi, version 6.2; Canonaco et al., 2013) to constrain the HOA, BBOA, Fac91, and IEPOX-SOA factors to be exactly the same **(i.e., using a-value of 0 in SoFi)** as the factor profiles found in unoxidized ambient air. **These four factors were not expected to be formed in the OFR, and were not observed to increase with OFR age in unconstrained runs. These factors were constrained so that we could calculate age-dependent changes in their mass concentrations using constant factor profiles (i.e., mass spectra), rather than allowing possible variations in factor profiles (with age) to confuse the interpretation of mass changes. The other two SOA-related factors were left unconstrained, in order to allow the analysis to determine the mass spectra of any SOA that was formed.**"

R2.6. Pg. 21 Lns. 7-9, Figure 10: The plateau of LO-OOA looks more of a slight increase because it was followed by a significant decrease. The LO-OOA was gradually increasing from around 0.5 eq. day, reaching a peak at around 2 eq. day, and then continuously decreasing.

We have changed the text at Pg. 21 Lns. 7-9 to clarify:

"**At lower ages, SOA associated with the LO-OOA factor was produced in increasing amounts with increasing age, peaking around approximately 2 eq. days of aging. At higher ages up to 6-9 eq. days, less mass formation associated with LO-OOA was observed.** Eventually at ages larger than 6-9 equivalent days a decrease of LO-OOA below the preexisting amount in ambient air was observed, indicating that the pre-existing ambient LO-OOA was being heterogeneously oxidized, and that no new LO-OOA was surviving the OFR (either it was not formed, or it was formed but then converted into MO-OOA or heterogeneously oxidized to gas-phase species)."

R2.7. Pg. 21 Lns. 16-17: It is unclear what the SOA from PMF analysis results is. It is because PMF analysis yielded LO-OOA and MO-OOA which can be referred as the SOA factor. In case of the dry season, IEPOX-SOA is also an SOA factor.

We have added the following text to the end of the sentence at Pg. 21 Ln. 17 to clarify:

" **(i.e., LO-OOA and MO-OOA)**".

As already noted at Pg. 21 Ln. 19, the mass spectrum of SOA formed in the OFR from various individual gases and from ambient air were too similar to be able to determine specifically which gases contributed to SOA from ambient air in the OFR. Thus, the PMF results alone do not provide further information on the sources of the SOA formed in the OFR.

Regarding IEPOX-SOA, it is indeed an SOA factor, but as noted at Pg. 20 Ln. 17 and extensively documented by Hu et al. (2016), IEPOX-SOA was not expected to be formed in the OFR due to experimental limitations.

R2.8. Pg. 23 Ln. 3: The authors refer to Thalman et al. (2017) for details analysis of ambient OA hygroscopicity (Korg). Thalman et al. (2017) reported Korg for particles with diameter range between ~90 to ~180 nm and showed that Korg was independent of particle size. Is there any reason for selecting Korg of 160 nm mobility diameter particles?

To address this comment, we have modified the sentence starting at Pg. 22 Ln. 14 to:

"Due to sampling time **and experimental** requirements, the **size-selected** CCN counter **sampled only a single particle mobility diameter during these measurements (160 nm), within the size range for which Thalman et al. (2017) had previously shown** $\kappa_{OA}$ **to be constant. Also,** these experiments were performed while keeping the **UV light intensity (and thus the approximate** amount of oxidation**)** in the OFR constant."

R2.9. Pg. 26 Lns. 11-14 and Pg. 27 Lns. 3-4: Here, the authors mention about multivariate relationship or multilinear regression (MLR). I think it would be good to provide the MLR model and coefficient values. Hence, it is clear how the SOA formation was predicted (Figure 12).

We have modified the following text at Pg. 26 Ln. 11 to clarify how the multilinear regression was calculated:

"If the assumption holds that VOCs and S/IVOCs from a given emission type correlate with each other, then a multivariate relationship should exist where the measured SOA formation should correlate with the sum of measured concentration of VOCs/tracers of each source multiplied by coefficients**:**

$$\textit{Measured SOA formation} = \Sigma(a_i \times c_i) \qquad\qquad (1)$$

where $c_i$ **is the concentration of the tracer for a given SOA source type, and** $a_i$ **is the coefficient for tracer** *i* **that leads to the best overall agreement with measured SOA formation.**"

Technical comments

R2.10. Pg. 4 Ln. 24: What does GoAmazon2014/15 stand for? Have the description of field campaign here rather than on Pg. 5 Ln. 5.

We have changed the text at Pg. 4 Ln. 24 from "GoAmazon2014/5" to **"Observations and Modeling of the Green Ocean Amazon (GoAmazon2014/5)"**, and made the opposite change at Pg. 5 Ln. 5.

R2.11. Pg. 5 Ln. 12: Add information that T2 site is downwind of Manaus. It is unclear here whether it is downwind or upwind.

Please see our response to R2.3.

R2.12. Pg. 6 Ln. 24: What is reference(s) for the typical 24 average ambient O3 concentration? Or is it from measurement at the site? Please clarify.

To clarify, the text at Pg. 6 Ln. 24 has been changed to:

"As with OH, the **value used for the typical $O_3$ mixing ratio (30 ppb) is meant to derive well-defined equivalent ages for a given exposure, as a guide for relative comparisons with other studies and sites. The** eq. age of $O_3$ oxidation can be scaled accordingly to apply a different average ambient $O_3$ concentration. **Average $O_3$ mixing ratios at the T3 site were 8 (19) ppb in the wet (dry) season, which would correspond to longer site and season-specific ages needed to reach a given $O_3$ exposure, according to the ratio of the $O_3$ mixing ratios.**"

R2.13. Pg. 7 Ln. 12: What does SQT stand for?

The text at Pg. 7 Ln. 12 now reads **"sesquiterpenes (**SQT**)"**, while the opposite change was made at Pg. 10 Ln. 17.

R2.14. Pg. 8 Ln. 12: Does it mean the ambient OA concentration is an average of measurements before and after OFR sampling? Please clarify.

The text has been changed at Pg. 8 Ln 12 to read:

"A key data product in this work is OA enhancement, which is defined as the OA concentration measured after oxidation minus the ambient OA concentration (**linearly interpolated from measurements** immediately before and after OFR sampling)."

R2.15. Pg. 10 Ln. 10: What does MT stand for?

The text at Pg. 10 Ln. 10 has been changed to **"monoterpenes (**MT**)"**, and the opposite change was made at Pg. 10 Ln. 17.

R2.16. Pg. 10 Ln. 17: The compounds and their acronyms should be provided earlier in the text. See technical comments #4 and 6.

These corrections were made in the responses to R2.13 and R2.15.

R2.17. Pg. 14 Ln. 7: What does LT mean?

The text has been changed from "LT" to **"local time (LT)"**.

R2.18. Pg. 17 Lns. 19-22: Which large chamber studies that OA mass concentrations are a factor of 2 comparable with the present study? Please add references. It may also be good to add the OA mass from previous chamber studies in Table S1 for comparison.

To address this comment, we have changed the text at Pg. 17 Ln 19 to:

"These yield values are generally consistent (within a factor of 2 for comparable OA mass concentrations) with the **Tsimpidi et al. (2010) and Henze and Seinfeld (2006)** values that **were** determined **using the results from** large chambers, with the averages being 0.9, 1.3, 0.5, and 0.9 times the respective chamber-derived yields for MT, SQT, toluene, and isoprene."

R2.19. Pg. 18 Ln. 11: Remove double from C=C bonds.

This change has been made.

R2.20. Pg. 20 Lns. 9-13: Add reference for IEPOX-SOA: Budisulistiorini et al. (2013) Add reference for Fac91: Robinson et al. (2011), Budisulistiorini et al. (2015)

We thank the reviewer for bringing the Fac91 references to our attention, and we have added the references on Pg. 20 Lns. 11-12. We have not included the additional IEPOX-SOA reference since multiple references are already included for that factor.

R2.21. Pg. 22 Lns. 21-25: Add reference(s) for observation of SOA-forming gases at the site (or nearby locations) during daytime and nighttime.

These observations were shown earlier in this manuscript. To clarify, we have changed the text at Pg. 22 Ln. 21 to read:

"During nighttime hours, when SOA-forming gases were **shown** to be present in ambient air in their highest amounts **(as discussed in Sects. 3.2, 3.3 and 3.5 above)**, the OFR was operated at a near constant age in the range of 1–3 eq. days. During daytime hours, when SOA-forming gases were present in lower concentrations **(as discussed in Sects. 3.2, 3.3 and 3.5 above)**, the OFR was operated at a near constant age in the range of 12–44 eq. days of OH aging.

R2.22. Figures 3 and 4: The unit of x axis (time) is better in Local Time (LT) because in the text you use LT.

We only used LT when showing or discussing diurnal cycles. For time series plots and discussion (including Figs. 3 and 4), we prefer to keep using UTC time.

R2.23. Figure 13: Are the labels (a) and (b) on figure caption switched? Please check.

We thank the reviewer for catching this mistake. We have changed the a) and b) in the Fig. 13 caption to read:

**"Fig. 13. a) The amounts and fraction of the total SOA formation from OH oxidation in the OFR at the T3 site that were attributed to biogenic, urban, and biomass burning emission types using multilinear regression analysis. b) Comparison of the average tracer concentrations and potential SOA formation during wet and dry seasons."**

Other Changes

1. Pg. 5 Ln. 27: "To investigate OH oxidation in the OFR, OH radicals were produced in situ using **mercury lamps with** the "OFR185" method described elsewhere (Li et al., 2015; Peng et al., 2015). **As in past field measurements with this OFR (Ortega et al., 2016; Palm et al., 2016, 2017), approximately 50 sccm of dry $N_2$ was constantly passed through each lamp sheath in order to prevent corrosion of the lamps and to reduce lamp-induced heating of the OFR."**

2. Pg. 29 Ln. 3: **"Also, higher ambient OA concentrations during the dry season were expected to lead to increased SOA yields (up to 2x larger) due to increased partitioning. This could explain a large fraction of the increased biogenic potential SOA, and would affect the potential from other sources as well."**

3. The measurement-based model of Pagonis et al. (2017) allows the estimation of the loss of SOA potential in inlets used to sample ambient air into an OFR. Since the loss of SOA potential to inlets is often a major problem in many OFR setups in the literature, we have added a brief mention in the manuscript and one figure to the Supp. Info. (Fig. S17) showing the results for a typical inlet, in order to alert others about the importance of minimizing inlets in OFR experiments.

Text added to main paper on Pg. 5 L 24: **"The avoidance of any inlet ahead of the OFR in this work was due to previous observations that showed a substantial decrease of SOA formation when using any inlets (Ortega et al., 2013). The model of Pagonis et al. (2017) allows for the first time a direct estimation of this effect. A case study using this model estimated that ~½ of the SOA potential would have been lost should a typical inlet have been used at the forested site of Palm et al. (2016) (Fig. S17). It is highly recommended that future studies also avoid the use of inlets ahead of an OFR, except when only very volatile precursors are used."**

[Figure]

**Fig. S17: Simulation of the effect of an inlet ahead of an OFR when measuring the SOA formation potential of ambient air. The case shown is based on the average SOA formation at the Manitou Forest site during the BEACHON-RoMBAS campaign, as described by Palm et al. (2016) and Hunter et al. (2017). It is assumed that a transient variation of the ambient SOA formation potential occurs at the field site, with a time scale of 10 min. In the case without an inlet (i.e., as performed in Palm et al., 2016), a total amount of SOA formation of 2.4 µg m$^{-3}$ would be observed without delay. In the case with a 10 m, ¼" OD, 2 lpm Teflon inlet (simulated with the model of Pagonis et al., 2017), the observed peak SOA formation is reduced by ~½, due to the very slow transmission through the inlet of species with c\* < 10$^5$ µg m$^{-3}$. Note that the residence time in the OFR is not considered in these simulations.**

[revised manuscript text omitted]